# Reconstructing Prehistoric Viral Genomes from Neanderthal Sequencing Data

**DOI:** 10.3390/v16060856

**Published:** 2024-05-27

**Authors:** Renata C. Ferreira, Gustavo V. Alves, Marcello Ramon, Fernando Antoneli, Marcelo R. S. Briones

**Affiliations:** 1Center for Medical Bioinformatics, Escola Paulista de Medicina, Federal University of São Paulo (UNIFESP), São Paulo, SP 04039-032, Brazilfernando.antoneli@unifesp.br (F.A.); 2Epigene LLC, São Paulo, SP 04537-080, Brazil; 3Computique LLC, São Paulo, SP 04545-006, Brazil

**Keywords:** Neanderthal genome, ancient viruses, sequence data, genome assembly, adenovirus, herpesvirus, papillomavirus

## Abstract

DNA viruses that produce persistent infections have been proposed as potential causes for the extinction of Neanderthals, and, therefore, the identification of viral genome remnants in Neanderthal sequence reads is an initial step to address this hypothesis. Here, as proof of concept, we searched for viral remnants in sequence reads of Neanderthal genome data by mapping to adenovirus, herpesvirus and papillomavirus, which are double-stranded DNA viruses that may establish lifelong latency and can produce persistent infections. The reconstructed ancient viral genomes of adenovirus, herpesvirus and papillomavirus revealed conserved segments, with nucleotide identity to extant viral genomes and variable regions in coding regions with substantial divergence to extant close relatives. Sequence reads mapped to extant viral genomes showed deamination patterns of ancient DNA, and these ancient viral genomes showed divergence consistent with the age of these samples (≈50,000 years) and viral evolutionary rates (10^−5^ to 10^−8^ substitutions/site/year). Analysis of random effects showed that the Neanderthal mapping to genomes of extant persistent viruses is above what is expected by random similarities of short reads. Also, negative control with a nonpersistent DNA virus does not yield statistically significant assemblies. This work demonstrates the feasibility of identifying viral genome remnants in archaeological samples with signal-to-noise assessment.

## 1. Introduction

Several hypotheses have been proposed to address the problem of Neanderthals’ extinction. These include: (I) Competition with modern humans, which suggests that modern humans outcompeted the Neanderthals for resources, such as food and shelter [1]. This could have resulted in the Neanderthals’ extinction, as they were less efficient in terms of surviving in the same environments as anatomically modern humans (AMH). (II) Interbreeding with modern humans. This hypothesis proposes that the Neanderthals were absorbed into the modern human population through interbreeding [2]. Over time, this hybridization could have led to the eventual disappearance of the Neanderthal lineage. (III) Environmental factors. Changes in the climate or other environmental factors, such as the onset of the last Ice Age, could have contributed to the Neanderthals’ decline and eventual extinction [3]. The Neanderthals may have been less adaptable to changing environmental conditions than AMH. (IV) Disease. It is possible that the Neanderthals were more susceptible to diseases carried by AMH [4]. This could have had a significant impact on their ability to survive and reproduce.

It is likely that a combination of these factors, along with other unknown factors, contributed to the eventual disappearance of Neanderthals. Among the proposed hypotheses, disease, and especially infectious diseases, could have played a relevant role in Neanderthal extinction [5]. Viral diseases caused by DNA viruses that produce persistent infections might leave genetic traces that can be detected in DNA from Neanderthal bone remains [6]. The hypothesis of Wolff and Greenwood [6] proposes that viruses that produce lifelong infections could have been introduced by modern humans among the Neanderthals, there is no evidence that these viruses were not already present in their common ancestor, or that it was Neanderthal populations that introduced these viruses into modern humans.

DNA viruses that are known to cause persistent infections in humans include: (I) Herpesviruses [7,8]. This family of viruses includes several members, such as herpes simplex virus (HSV), varicella-zoster virus (VZV), and Epstein–Barr virus (EBV). These viruses can establish lifelong infections in their hosts and remain dormant for prolonged periods of time, periodically reactivating to cause recurrent disease. (II) Papillomaviruses [9]. These viruses are responsible for causing human papillomavirus (HPV) infections, which can lead to the development of several types of cancers, including cervical cancer. The virus can persist in the body for many years, even decades, without causing any symptoms. (III) Polyomaviruses [10]. This family of viruses includes Human polyomavirus 2, commonly referred to as the JC virus or John Cunningham virus, and the BK virus, also known as human polyomavirus 1, which can cause persistent infections in the kidneys and other organs. These viruses are often reactivated in people with weakened immune systems, such as transplant recipients. (IV) Adenoviruses [11]. Human adenoviruses (HAdV) can cause persistent infections in the tonsils, adenoids, and other mucosal tissues. These viruses can also cause respiratory and gastrointestinal infections. (V) Hepatitis B virus (HBV) [12]. This virus can cause chronic hepatitis, which can lead to cirrhosis and liver cancer. It is estimated that around 250 million people worldwide are chronically infected with HBV.

The persistence of these viruses is dependent on their ability to evade the immune system and/or establish latency, where the virus becomes dormant within the host cells [13]. This makes it difficult for the host to eliminate the virus and can lead to chronic infections.

Paleogenomics provide insight into the evolution of transmissible and nontransmissible diseases [8,14]. DNA samples from bones of human ancestors are a source of genomic information of the hosts as well as pathogens [15]. Ancient viral genomes more likely to be identified in Neanderthal raw sequence data might be DNA viruses causing persistent infections [16]. Among DNA viruses causing persistent, or long-term, infections are herpesviruses, papillomaviruses, and adenoviruses. Herpesviruses, in particular, might have been a major cause for Neanderthal extinction, as proposed in previous studies [4,6,17]. Recovery of herpesvirus DNA and microRNAs from nonprimary infections has been reported from the teeth and subgingival plaque of living individuals [18,19]. HSV1 DNA has been recovered from the trigeminal ganglia (TG) of cadavers, indicating that it can reactivate peri- or post mortem [20]. Although herpesvirus (HSV1) DNA is detectable in blood during primary infection, it is not detected in reactivation from latency [21]. Herpesvirus (European HSV1) has been successfully recovered from samples dating from the 3rd to 17th century CE, with up to 9.5× coverage with paired human genomes up to 10.16× coverage [8]. Therefore, herpesvirus must be common in the archaeological record.

In this study, we addressed the hypothesis of Neanderthal infection with adenovirus, herpesvirus and papillomavirus by testing whether it is possible to identify remnants of viral genomes in Neanderthal sequencing data. As proof of concept, we searched for viral genomes in raw sequencing data of *Homo sapiens neanderthalensis* from Chagyrskaya cave in Russia (samples 1 and 7) [22]. Analysis of taxonomic output of the Sequence Taxonomic Analysis Tool (STAT) on the NCBI Sequence Read Archive (SRA) [23] revealed that reads corresponding to several microorganisms, such as bacteria, fungi and viruses, were detected in these samples. Inspection of the results from this analysis allowed for the selection of genomes of human pathogenic DNA viruses to be used as references (templates) for genome assemblies.

## 2. Materials and Methods

### 2.1. Samples, Representative Sequences, and Reference Sequences

Neanderthal reads were downloaded from the Sequence Read Archive (SRA) at the National Center for Biotechnology Information (NCBI) under accession numbers PRJEB55327, Runs ERR10073004 to ERR10073014 for Neanderthal Chagyrskaya 1 (from lower left deciduous canine) and Runs ERR10073054 to ERR10073181 for Neanderthal Chagyrskaya 7 (from thoracal vertebral process fragment). Representative sequences, used as assembly references, were from human adenovirus 7 (HAdV-7) (KX897164), human alphaherpesvirus 1 (HSV1) (MN136523) and human papillomavirus 12 (HPV12) (X74466). The human genome reference sequence used was from BioProject PRJNA31257, NCBI RefSeq assembly GCF_000001405.40 and GenBank assembly GCA_000001405.29 (https://www.ncbi.nlm.nih.gov/grc accessed on 7 February 2024). Reference sequences (NCBI RefSeq https://www.ncbi.nlm.nih.gov/refseq/ accessed on 27 November 2023 and ICTV https://ictv.global/ accessed on 20 March 2024) used for the analyses were from human adenovirus: AC000007.1, AC000008.1, NC012959.1, NC001460.1, NC011203.1, NC011202.1, NC001405.1, AC000006.1, NC010956.1, NC003266.2, NC001454.1, AC000017.1, AC000018.1, AC000019.1, DQ086466.1, AY163756.1, AY803294.1, AY601636.1, AY601633.1, AY737797.1, AY128640.2 and AY737798.1. Those from simian adenovirus were FJ025930.1, HI964271.1, FJ025910.1, HC085083.1, HC085052.1, HC085020.1, FJ025915.1, HC084988.1 and AR101858.1. Those from human herpesvirus were NC001806.2, NC001798.2, NC001348.1, NC009334.1, NC001716.2 and NC009333.1. Those from human papillomavirus were NC001676.1, NC013035.1, NC001356.1, NC001352.1, NC001357.1, NC001694.1, NC027779.1, NC026946.1, NC001457.1, NC001595.1, NC001596.1, NC001576.1, NC001526.4, NC001583.1, NC001586.1, NC001587.1, NC001354.1, NC001690.1, NC001591.1, NC001691.1, NC001593.1, NC001693.1, NC001458.1, NC034616.1, NC010329.1, NC004104.1, NC004500.1, NC005134.2, NC008189.1, NC008188.1, NC012213.1, NC012485.1, NC012486.1, NC014185.1, NC014955.1, NC017993.1, NC017994.1, NC017995.1, NC017996.1, NC017997.1, NC028125.1, NC022892.1, NC023891.1, NC027528.1 and NC001531.1. For the DNA virus negative control, the human parvovirus B19 reference sequence used was NC000883.2. The primate (non-human) and murid NCBI RefSeq viral sequences used were Chimpanzee adenovirus Y25 NC017825.1, Murine adenovirus 3 NC012584.1, Murine adenovirus A NC000942.1, Chimpanzee alphaherpesvirus strain 105640 (panine alphaherpesvirus 3) NC023677.1, Macacine herpesvirus 1 NC004812.1, Murid herpesvirus 1 NC004065.1, *Mus musculus* papillomavirus type 1 isolate MusPV NC014326.1, *Rattus norvegicus* papillomavirus 3 isolate Rat_60S NC028492.1 and rhesus macaque papillomavirus NC001678.1.

### 2.2. Genome Assembly by Reference Mapping

For genome assembly by mapping, duplicate reads were removed using Dedupe Duplicate Read Remover 38.84 and trimmed for illumina TrueSeq adapter/primers using BBDuk. The resulting reads were mapped to viral assembly reference sequences (templates) and NCBI RefSeq sequences using BBMap with k-mer length = 13, random map of multiple best matches, maximal indel size = 16,000 and normal sensitivity [24,25] (https://sourceforge.net/projects/bbmap/ accessed on 20 November 2022). Consensus sequences were obtained using the strict 50% majority rule and high-quality 60%. BBMap, a splice-aware global aligner for DNA and RNA sequence reads, outperforms Bowtie 2 and BWA for short read sequences because it is fast and extremely accurate, particularly with highly mutated genomes or mapping that produce long indels (up to 100 kb long) [25]. Images were processed using Geneious Prime 2023 (https://www.geneious.com accessed on 10 March 2023).

### 2.3. Tracking and Quantification of DNA Damage Patterns in DNA Sequence Reads

The ancient DNA origin of Neanderthal viral reads was tested in BAM files using a Bayesian method implemented in mapDamage, with multiple time runs and convergence tests [26]. In each of the three assemblies, 50,000 iterations were run, and the acceptance ratio was in the admissible range of (0.1–0.3) for all estimated parameters. MapDamage was run as a docker via “docker container run -it quay.io/biocontainers/mapdamage2:2.2.0--pyr36_1 /bin/bash” to create the environment. The following command was used, inside the created environment, to run the program and define the plot title: “mapDamage -i mymap.bam -r myreference.fasta -t title”. Adenovirus reads used for the negative control of mapDamage analysis were from NCBI SRA SRX22331168: whole-genome sequencing of human adenovirus sp., accession PRJNA930027 (run SRR26630751) containing 11,833,726 reads and 1.4 G bases (illumina NextSeq 2000). The positive control for mapDamage analysis consisted of Neanderthal reads of Chagyrskaya 7 mapped to the human mitochondrial genome reference rCRS (NC_012920.1).

### 2.4. Viral Genealogies

The consensus sequences obtained from genome assembly by reference mapping were exported and used for a GenBank wide blast search (MegaBlast) to identify their closest relatives among extant viruses [27]. Alignments of consensus sequences (queries) with closest relatives were obtained by MegaBlast and genealogies by neighbor-joining as implemented in Blast with default parameters [28]. Additionally, viral genealogies were inferred using maximum likelihood as implemented in PAUP 4* (https://paup.phylosolutions.com/get-paup/ accessed on 10 March 2023) [29] and the inference of Bayesian trees as implemented in MrBayes 3.2.7a (https://nbisweden.github.io/MrBayes/index.html accessed on 10 March 2023) [30]. In maximum likelihood inference, the initial model parameters were obtained via ModelTest-NG 0.1.7 using the Bayesian Information Criterion (BIC) [31] as implemented in CIPRES (https://www.phylo.org/ accessed on 23 October 2023) [32]. For Bayesian trees inference, the model parameters (base frequencies, substitution rates, proportion of invariants and gamma distribution shape) were estimated during hot and cold chain tree search by MCMC with a burn-in length of 110,000, subsample frequency of 200 and total chain length of 1,100,000. All phylogenies were inferred from alignments produced with the MAFFT program (version 7.490) with automatic selection for the algorithms L-INS-i, FFT-NS-i and FFT-NS-2 according to data size, scoring matrix 200 PAM/k = 2, gap penalty = 1.53 and offset value = 0.123 [33]. All alignments and trees were inferred from whole-genome sequences. Phylogenies of HAdV-7-N1, HSV1-N1 and HPV12-N1 compared to NCBI RefSeq non-primate, and murid sequences were inferred using a maximum likelihood algorithm as implemented in FastTree 2.1.11 with the GTR model, 4 categories of substitution rates and branch support by the Shimodaira–Hasegawa test [34]. Trees with the closest extant relatives of Neanderthal viruses, RefSeq sequences and non-human primate sequences were inferred using the Bayesian method as implemented in MrBayes 3.2.7 with total chain length = 1,100,000, subsample frequency = 200 and burn-in length = 110,000.

### 2.5. Assemblies with Random Reference Sequences

Random mock reference sequences of adenovirus, herpesvirus and papillomavirus were obtained using the random sequence generator of UGENE program version 44 [35]. Random mock sequences were of the same size and base frequencies as actual assembly reference sequences. Random reference sequences were used for genome assembly by mapping using the same programs and parameters used for actual reference sequences, as described above. A comparison of assemblies by mapping of real and random reference sequences was assessed using Welch’s *t*-test, a modification of Student’s *t*-test which does not assume that the two samples have equal variances, as is the case with the data considered herein [36].

## 3. Results

### 3.1. Taxonomic Assessment of Neanderthal Raw Genomic Data

An initial taxonomic analysis of sequence reads, based on MinHash-based k-mers, (STAT, https://www.ncbi.nlm.nih.gov/sra accessed on 15 October 2022) [23], revealed that reads of human pathogenic DNA viruses such as herpesviruses, papillomaviruses and adenoviruses were present in Neanderthal sequencing runs in ≈0.01% reads (Chagyrskaya 1 and 7) (Table 1) [22]. The examination of the analysis, available on the SRA site, for Chagyrskaya 1, comprised 11 runs with 2.96 × 10^9^ bases (3.25 Gb of data). To test whether the reads were viral in the SRA analysis of Neanderthal reads, we analyzed these reads in more detail by mapping to extant viral genomes.

### 3.2. Genome Assembly by Mapping

Because of the higher coverage of Neanderthal genome Chagyrskaya 7 (4.9-fold coverage) [22], it was used for reference mapping using BBMap, an assembler optimized for short reads. Before processing, the data comprised 128 runs with 152.71 × 10^9^ bases (170.21 Gb of data). After removal of duplicate reads, trimming and size selection (exclusion of reads < 10 bases), a total of 1,221,449,544 reads of Neanderthal Chagyrskaya 7 were mapped simultaneously to representative genomes of adenovirus (180,420 reads), herpesvirus (1,224,414 reads) and papillomavirus (23,999 reads) (Figure 1). A total of 1,220,020,414 reads did not map to the references. In simultaneous mapping with different references, each reference worked as a decoy for the other assemblies.

In Figure 1a, the assembly of Neanderthal reads to adenovirus has a mean read length of 17 bp, with 3090 (8.0%) identical sites and pairwise nucleotide identity of 86.5%. The coverage of 38,528 bases is 102.2 (mean), with a standard deviation of 249.4. The maximum coverage is 3802 with confidence mean of 40.8, expected errors of 2374, error-free odds < 0.0001%, Q20 = 99.6%, Q30 = 99.4% and Q40 = 98.7%. The exported Neanderthal consensus sequence was named HAdV-7-N1 (human adenovirus 7—Neanderthal 1).

In Figure 1b, the assembly of Neanderthal reads to herpesvirus has a mean read length of 18 bp, with 5383 (2.9%) identical sites and pairwise nucleotide identity of 81.6%. The coverage of 185,422 bases is 171.2 (mean), with a standard deviation of 247.4. The maximum coverage is 6771 with a confidence mean of 40.8, expected errors of 15,416, error-free odds < 0.0001%, Q20 = 99.6%, Q30 = 99.4% and Q40 = 98.7%. The exported Neanderthal consensus sequence was named HSV1-N1 (human herpesvirus 1—Neanderthal 1).

In Figure 1c, the assembly of Neanderthal reads to papillomavirus has a mean read length of 20 bp, with 1069 (13.0%) identical sites and pairwise nucleotide identity of 85.5%. The coverage of 8223 bases is 115.4, with a standard deviation of 609.1. The maximum coverage is 6401 with a confidence mean of 40.8, expected errors of 378.97, error-free odds < 0.0001%, Q20 = 99.6%, Q30 = 99.4% and Q40 = 98.8%. The exported Neanderthal consensus sequence was named HPV12-N1 (human papillomavirus 12—Neanderthal 1).

The average number of read lengths that mapped to viral reference genomes was between 17 and 20 bases, and the mapping was consistent in the three viral assemblies (Figure 2, Figure 3 and Figure 4). In the adenovirus assembly (Figure 1a), the distribution of read lengths ranged from 13 to 139 bases (Figure 5a). In the herpesvirus assembly (Figure 1b), the read length distribution ranged from 13 to 93 bases (Figure 5b). In the papillomavirus assembly (Figure 1c), the read length distribution ranged from 13 to 61 bases (Figure 5c).

### 3.3. Analysis of Deamination Patterns in Neanderthal Genomic Reads

To show that the reads mapped to viral genomes were of aDNA origin and not contamination with modern DNA, the reads mapped to viral genomes were tested via analysis of deamination as implemented in the program mapDamage (Figure 6), which used the BAM files of assemblies shown in Figure 1 as inputs. In Figure 6a,c, the assembly of adenovirus and papillomavirus shows that C-to-T deamination was significantly more abundant in the 5′ overhangs as compared to the other changes away from termini. In the herpesvirus assembly, although C-to-T deamination is more prevalent in the termini, C-to-T is also more abundant in central regions than other changes (Figure 6b). The likelihood provides statistical support for C-to-T deamination in all three assemblies, which is a main feature of reads derived from aDNA as opposed to recent DNA [26]. The G-to-A, as observed in Figure 6, could be caused by deamination of guanine residues to xanthine (X) residues, causing the basecall of adenine residues (A) in DNA sequencing and leading the observed G-to-A pattern in terminal residues in aDNA reads (Figure 6). For the mapDamage analysis, a positive and a negative control were included. The positive control consisted of Neanderthal reads aligned to the modern human mitochondrial genome (mtDNA). This is to show the deamination pattern with bona fide human aDNA reads mapped to the reference. The negative control consisted of reads from a present-day human adenovirus 7 genome project (PRJNA930027, run SRR26630751) aligned to the human adenovirus 7 reference sequence, as this was supposed not to have the aDNA deamination pattern. These controls are depicted in Figure 6d,e.

### 3.4. Genome Mapping with Taxonomic Reference Sequences

To verify that the assemblies in Figure 1 were supported by assemblies with viral reference sequences, Neanderthal reads were also mapped to reference viral genomes of adenovirus (14 NCBI RefSeq sequences), herpesvirus (6 NCBI RefSeq sequences) and papillomavirus (45 NCBI RefSeq sequences) to test the preferential mapping of Neanderthal reads to reference sequences (Table 2). The analysis with taxonomy reference sequences showed that the two adenovirus reference sequences with the most reads mapped were AC_000018.1 (HAdV-7 with 78,255 reads) and NC_011203.1 (HAdV-B1 with 54,835 reads); for herpesvirus, the two best matches were NC_001806.2 (HSV1 strain 17 with 937,258 reads) and NC_001798.2 (HSV2 strain HG52 with 167,080 reads); and for papillomavirus, the two best matches were NC_001531.1 (HPV5 with 1889 reads) and NC_001591.1 (HPV49 with 494 reads) (Table 2).

### 3.5. Genetic Relatedness of Neanderthal Viruses and Extant Viruses

To verify the relatedness of the inferred Neanderthal viral sequences to their closest extant relatives in a wide database, the consensus from mapping assemblies was exported and used in a MegaBlast search of the GenBank. Each consensus sequence from the three assemblies in Figure 1 (Appendix A) was used as a query. The closest extant relatives identified in the search are depicted as neighbor-joining trees (Appendix A). In Appendix A, the Neanderthal adenovirus is distant from any extant adenovirus genome by at least 0.06 substitutions/site, which corresponds to 2312 substitutions (100 closest relatives) (alignment in Appendix A). In Appendix A, the Neanderthal herpesvirus is 0.07 substitutions/site, which corresponds to 12,780 substitutions along ≈50,000 years. The Neanderthal herpesvirus is very distant from any present-day herpesvirus genome (709 closest relatives) (alignment in Appendix A). In Appendix A, the Neanderthal papillomavirus clusters has sequences of type 12, although its branch length indicates its distant relationship to other sequences in the same cluster at 0.06 substitutions/site or 494 substitutions along the branch (72 closest relatives) (alignment in Appendix A).

From the MegaBlast alignments, the ten closest relatives of each Neanderthal virus were extracted for phylogenetic inference using the maximum likelihood method (Appendix A). For papillomavirus alignment, only the nine closest relatives were included due to the exclusion of a partial sequence (MH777358.2). These phylogenies were consistent with relatedness inferred by MegaBlast (Appendix A) and were inferred from alignments in Appendix A. The starting parameters of maximum likelihood trees were estimated by ModelTest-NG using the Bayesian Information Criterion (BIC) (Appendix A). Bootstrap analysis showed that the Neanderthal sequences were separated from their closest extant relatives with 100% frequency (Appendix A).

The consensus sequences of HAdV-7-N1, HSV1-N1 and HPV12-N1 reconstructed here are included in the Appendix A, and the differences between the reconstructed Neanderthal sequences and the respective assembly reference sequences are given in Appendix A (VCF files are in Appendix A).

The closest relatives in the phylogenies shown in Appendix A were used, along with the corresponding NCBI RefSeq sequences shown in Table 2 for each corresponding virus family plus a non-human primate sequence (chimpanzee for adenovirus and herpesvirus and rhesus macaque for papillomavirus). The corresponding alignments for phylogenetic inference are given in Appendix A. The Bayesian trees (Figure 7) show that the closest relative to HAdV-7-N1 (excluding the assembly reference sequence) was KP670856.2 (HAdV-7 strain human/CHN/GZ6965/2011/7[P7H7F7]), with 94.9% nucleotide identity. The closest relative to HSV1-N1 (excluding the assembly reference sequence) was KX265031.1 (HSV1 strain 914-H2), with 93.8%. The closest relative of HPV12-N1 (excluding the assembly reference sequence) was MH777286.2 (Metagenome Assembled Genome: human papillomavirus isolate HPV-mSK_144), with 82.3% (Appendix A).

The relationships of HAdV-7-N1, HSV1-N1 and HPV12-N1 with RefSeq non-human primate viral sequences were assessed according to phylogenies (Appendix A). The branching patterns followed the expected distances according to the divergence of the hosts. Also, as expected, the Neanderthal viral sequences clustered together with modern human viral sequences and were distant from the chimpanzee sequences (there were no full-length papillomavirus chimpanzee genomes in RefSeq, and rhesus macaque papillomavirus genome NC_001678.1 was used instead). Nucleotide identity matrices (Appendix A) showed that HAdV-7-N1 and HSV1-N1 were ≈95% identical to human sequences and ≈70% identical to chimpanzee sequences, while HPV12-N1 was 60–75% identical to the human viral counterparts and ≈40% identical to the rhesus macaque counterpart (Appendix A).

Regarding the possible degeneration of codons due to ambiguities generated by deamination biases, it must be noted that the exclusion sets in phylogenetic analysis remove characters (positions) with IUPAC ambiguity codes. This degeneration would affect only C-to-T and G-to-A transitions. Even if the C-to-T deamination were so high as to generate a position with an IUPAC ambiguity code, it would be removed in the phylogenetic analysis. In the HAdV-7-N1 alignment (34,662 positions), the excluded characters were 16 gapped, 1526 missing/ambiguous and 32,235 non-gapped invariants, leaving a total of 885 positions from which the trees were inferred. In the HSV1-N1 alignment (150,608 positions), the excluded characters were 9880 gapped, 7417 missing/ambiguous and 128,255 non-gapped invariants, leaving a total of 5056 positions from which the trees were inferred. In the HSV1-N1 alignment (7538 positions), the excluded characters were 934 gapped, 254 missing/ambiguous and 3869 non-gapped invariants, leaving a total of 2481 positions from which the trees were inferred. The C-to-T and G-to-A deamination did not occur in all Cs or all Gs at termini, and therefore were not confounded with a bona fide C-to-T or G-to-A substitution, which was called in the consensus when above 50% or with a high-quality (phred score ≥ 30) position (Appendix A).

### 3.6. Putative Changes in Proteins of Neanderthal Viruses

Alignments of the inferred HAdV-7-N1, HSV1-N1 and HPV12-N1 to the closest extant relatives revealed the sequence changes distributed along most coding regions, including those encoding relevant infection proteins (Figure 8). In adenovirus, changes were observed in genes encoding Fiber (L5), Hexon (L3) and Penton base and core proteins (L2) (Appendix A) [37]. In herpesvirus, changes were observed in glycoproteins involved in interaction with host cell surfaces such as gB (UL27), gD (US6), gH (UL22) and gL (UL1) (Appendix A) [38]. In papillomavirus, changes were observed in capsid protein L1, as well as oncoproteins E6 and E7 (Appendix A) [39].

The non-synonymous SNPs detected in Neanderthal viral genes showed that the effect of deamination was minimal, because most C-to-T and G-to-A deamination changes were below the threshold for consensus sequence generation. In other words, the C-to-T and G-to-A changes were located preferentially in the termini of individual reads in quantities and patterns compatible with aDNA, as shown above, but not all reads had these changes. Therefore, although a deamination pattern was present, it was below 50% frequency, as shown in Appendix A. For this reason, the deamination bias did not have to be compensated by phylogenetic analysis and did not compromise the inference of amino acid changes from viral DNA sequences. In the case of C-to-T changes that lead to amino acid changes, the frequency of the T in a “C” position has to be higher than the frequency expected by mere deamination alone, and therefore, the mere deamination can be differentiated from a bona fide C-to-T transition.

### 3.7. Assessment of Random Effects in Genome Mapping

To assess whether mapping of Neanderthal reads to viral references could be obtained by chance alone, due to short reads, random viral references sequences were generated and mapped (Figure 9). Random references have the same size and base compositions as the actual references, namely, human adenovirus 7 (KX897164), human alphaherpesvirus 1 (MN136523) and human papillomavirus type 12 (X74466). The idea is to observe the statistics of Neanderthal reads mapped to actual viral sequences versus their random versions (Table 3). In Figure 9a, the assembly of Neanderthal reads to the random adenovirus reference has a coverage of 62.2 (mean) and a standard deviation of 180.2, with 126,613 reads mapped. In Figure 9b, the assembly of Neanderthal reads to random herpesvirus reference has a mean coverage of 154.8 and a standard deviation of 704.3, with 1,166,326 reads mapped. In Figure 9c, the assembly of Neanderthal reads to random papillomavirus reference has a mean coverage of 51.0 and a standard deviation of 116.0, with 22,682 reads mapped (Table 3). Welch’s *t*-test was used to assess whether mapping of Neanderthal reads with real templates (assembly references) produced assemblies with higher coverage as compared to random reference sequences (Table 3). Comparison of the real *t*-statistics references of human adenovirus 7 (KX897164), human alphaherpesvirus 1 (MN136523) and human papillomavirus type 12 (X74466) and random references showed that the real template assembly coverages were always above the random reference assemblies (the *t* value is always above the critical value for infinite degrees of freedom because of the high number of reads), with *p*-values = 0 (probability zero of rejecting a correct null hypothesis).

To further test for artifactual mapping of Neanderthal reads on viral references, the mapped reads in Figure 1 were extracted (“unassembled”) from the bam files and used for mapping to the human genome scaffolds (hg38). No reads mapped to human nuclear chromosomes and only a limited number of reads mapped to human mitochondrial DNA (Figure 10), which shows that reads that map to viral genomes do not map to the human genome, and therefore, random similarities due to short reads do not explain the results observed in Figure 1, *per se*. The 180,419 HAdV-7-N1 reads were assembled to hg38, and 140 reads mapped to mtDNA while 180,279 reads were not assembled to any human chromosome. The 1,224,713 HSV1-N1 reads were assembled to hg38, and 465 reads mapped to mtDNA while 1,224,248 reads were not assembled to any human chromosome. Finally, the 23,998 HPV12-N1 reads were assembled to hg38, and 16 reads mapped to mtDNA while 23,982 reads were not assembled to any human chromosome.

The same set of unassembled reads was used for assembly by mapping to the corresponding random reference sequences to test whether simple, random similarities could account for the mapping coverage—in other words, if random noise could explain the results (Figure 11). The analysis revealed that 232 of 180,419 HAdV-7-N1 reads were assembled to the adenovirus random reference, while 180,187 reads were not assembled (Figure 11a), which is the expected result if mapping is not due to random similarities due to the short size of reads. A total of 84,062 of the 1,224,713 HSV1-N1 reads were assembled to the herpesvirus random reference, while 1,140,651 reads were not assembled (Figure 11b), and 1 of 23,998 reads was assembled to the papillomavirus random reference, while 23,997 reads were not assembled (Figure 11c).

Taken together, the results of the controls in Figure 9, Figure 10 and Figure 11 show that pure randomness does not explain the results observed in Figure 1. As shown in Figure 9, we tested the level of coverage when Neanderthal reads were mapped to random mock sequences and compared the coverage level with real assembly references. As shown in Figure 10, we took the reads mapped to viral genomes and mapped them to the human genome (hg38). The reads that mapped to random references were used to map to the human genome (hg38) (Figure 11). The analysis in Figure 9 shows that the coverage of mapping to viral genomes was higher than with random references. The test in Figure 10 shows that none of the reads that mapped to real viral genomes mapped to human nuclear sequences, and only a few reads mapped to mitochondrial DNA. In Figure 11, the analysis reveals that the reads that mapped to random viral references did not map to the human nuclear genome, and very few sequences mapped to portions of the mitochondrial DNA. Therefore, random effects due to artifactual similarities, because of the read lengths used herein, did not explain the mapping of Neanderthal reads to viral genomes used for mapping, *per se* (Figure 1).

### 3.8. Control with a Nonpersistent DNA Virus

To test whether a nonpersistent DNA virus would show a positive result with Neanderthal reads, we mapped the Neanderthal reads against the human parvovirus B19 reference sequence (NC_000883.2) [40]. Parvovirus B19 has a DNA genome, only infects humans and is known for causing disease in the pediatric population, as well as rarely in adults. It is the cause of the childhood rash called fifth disease, also known as erythema *infectiosum* or “slapped cheek syndrome” [41]. Individuals with anti-parvovirus B19 IgG antibodies are generally considered immune to recurrent infection [42]. The assembly was carried out with the same set of Neanderthal reads used for mapping of adenovirus, herpesvirus and papillomavirus shown in Figure 1. To estimate the signal-to-noise level, mapping against parvovirus also included three random mock sequences whose lengths and percentages of bases A, C, G and T were identical to those of the parvovirus reference sequence. By signal, we mean the mapping coverage and distribution against the parvovirus reference, and by noise, we mean the coverage and distribution against random references. A positive result was considered when the mapping statistics were above those observed by using random references, while a negative result was considered when the mapping statistics were equivalent to those observed for random references. The assembly by mapping with parvovirus (Figure 12) showed that 714 Neanderthal reads mapped to the parvovirus reference with a mean coverage of 2×, standard deviation of 9.2 and 100% single strand (Figure 12a). The mapping of Neanderthal reads against random reference sequences 1, 2 and 3 (Appendix A) (Figure 12b–d) showed mean coverages of 3.5× (1271 reads, SD = 16.9), 7.2× (2595 reads, SD = 41.6) and 2.7× (975 reads, SD = 10). The distribution of reads along the references also showed that the pattern observed for the parvovirus was equivalent to the pattern for random references, and therefore, the result was negative; in other words, parvovirus DNA could not be detected in Neanderthal reads above the level expected by random noise. Welch’s *t*-test was used to determine whether mapping of Neanderthal reads with human parvovirus B19 produced assemblies with higher coverage as compared to random reference sequences (Table 4). Comparison of the *t* statistics of parvovirus B19 and random references showed that parvovirus B19 assemblies were always below (random Appendix A) or equal to random references (random Appendix A) with proper *p*-values. Therefore, the qualitative pattern observed in Figure 12 is statistically supported (Table 4). When parvovirus B19 was tested against random Appendix A, the coverage with random references was statistically higher than with the real reference, while testing against Appendix A showed that mapping coverage with random reference was statistically identical to that with the real reference. In all three cases, the coverage with the real reference was never statistically higher than with the random references, and therefore, the result was negative. In other words, the null hypothesis of randomness cannot be refuted.

### 3.9. Herpesvirus Type 1 (HSV1) and Type 2 (HSV2)

Regarding the detection of herpesvirus types in Neanderthal genome data, the initial STAT binning analysis suggested that HSV2 was the type detected in Chagyrskaya 7 (Table 1), while our genome mapping used an assembly reference sequence of HSV1 that was detected in Chagyrskaya 1 (Table 1). To test whether HSV1 and HSV2 could be discriminated with assembly by mapping, a total of 1,221,442,735 Neanderthal Chagyrskaya 7 reads were assembled by mapping using reference sequences of HSV1 (NC_001806.2) and HSV2 (NC_001798.2) (Figure 13). A total of 1,221,263,284 reads were not assembled to either viral reference. The HSV1 mapping produced a contig with a mean coverage of 8.1× (SD = 19, Q40 = 98.6%) and 74,405 reads, while HSV2 produced a contig with a mean coverage of 11.4x (SD = 29.2, Q40 = 98.6%) and 105,046 reads. The alignment of reference genomes for HSV1 and HSV2 showed that the global sequence identity was 73%, with 80% percent in the most conserved domains and 50% in the most divergent regions (Appendix A). Given the read lengths and algorithms used, it is possible that reads mapping to the conserved regions might have been responsible for identifying either HSV1 or HSV2 in Neanderthal samples. Another possibility is that mixed infections with HSV1 and HSV2 can explain these results. Mixed infections with HSV1 and HSV2 occur in 6–11% of HSV infections, according to PCR testing with type-specific primers [43,44].

## 4. Discussion

In the current study, we present an initial assessment of the possible presence of viral DNA remnants in Neanderthal sequencing data. For this purpose, we analyzed the SRASTAT results of Neandertal reads of the Chagyrskaya Cave. This binning method indicated that approximately 0.01% of Neanderthal reads, more specifically, Chagyrskaya 1 and 7, contained potential viral sequences. To investigate this possibility in more detail and to determine the exact number of viral reads in these data, we mapped Neanderthal reads to viral reference sequences. The initial SRA analysis indicated that double-stranded DNA viruses that produce lifelong infections could be present in these Neanderthal samples, as previously hypothesized [6]. Mapping of Neanderthal reads to adenovirus, herpesvirus and papillomavirus reference genomes produced consistent assemblies by mapping, and the inferred consensus sequences were analyzed and compared to the closest extant relatives. We show that changes in Neanderthal viruses are distributed along the sequences, and nonsynonymous changes are observed in coding regions of surface proteins such as glycoproteins gB (gene UL27), gD (gene US6), gH (gene UL22) and gL (gene UL1), which are involved in cell adhesion and infectivity of herpesviruses (Appendix A) [45]. Changes in the major viral surface proteins of adenovirus (genes L2, L3 and L5) and oncoproteins of papillomavirus (genes E6 and E7) were revealed as well (Figure 8, Appendix A). A detailed analysis of these amino acid changes and their putative functional effects on these proteins is a matter for future research. Most positions of Neanderthal viruses inferred herein were conserved, and therefore, several segments surrounding variable regions and regions with low coverage in mapping can be used as targets for PCR primer design. The amplicons obtained through PCR from purified Neanderthal DNA can be sequenced to fill low-coverage regions, which would allow for the full reconstruction of these paleoviruses and their Neanderthal epidemiology. Further detailed characterization of variable regions of these ancient viral genomes might reveal adaptive Neanderthal signatures.

In the case of Chagyrskaya 7, because DNA was isolated from a thoracal vertebral process, it is likely that DNA from a latent viral infection, especially herpesvirus, was coisolated with Neanderthal DNA. HSV1 and HSV2 are part of the alpha subfamily of human herpesviruses. The lytic phase of infection occurs within mucoepithelial cells, and latent infection occurs in neurons. HSV1 establishes latency preferentially in sensory neurons of peripheral ganglia and is associated with orofacial infection. While HSV1 is isolated from the upper half of the body, innervated by the trigeminal ganglia, HSV2 is isolated from the lower half of the body innervated by the sacral ganglia, which explains the clinical pattern associated with genital herpes infection [43]. However, recent studies indicate that this viral tropism is not always applicable and mixed infections occur in both ocular and genital herpes patients [43]. Human adenovirus persistence and reactivation occurs in the gastrointestinal tract, and the best-documented sites of latency include tonsillar and adenoidal T lymphocytes [46]. The latency of papillomavirus is associated with the basal epithelial stem cell pool and is periodically induced to reactivate when the stem cell divides, and one daughter cell is committed to terminal differentiation and induction of the viral life cycle. Tissue-resident memory T-cells are hypothesized to control these periodic reactivation episodes and thus limit their duration [47]. Taken together, these data suggest that, once infected by these viruses, the Neanderthal host might have remained with viral particles in its body tissues for its whole life. Depending on viral loads, it is likely that sufficient viral DNA remained *postmortem* and was coisolated and sequenced. Although it is relevant to determine whether Neanderthal individuals were infected with HSV1 and HSV2, it seems that, with the methodology used in this study, it is not possible to give a definitive answer, because Neanderthal reads could map with approximate probability, depending on the binning algorithm and parameters, to conserved regions (Appendix A). Therefore, a deeper specific analysis should be carried in future studies. Also, mixed infections with HSV1 and HSV2 are not uncommon (approximately 10% patients), and these can be detected in samples from patients via PCR amplification using HSV1/HSV2-specific primers [43,44]. Our result shown in Figure 13 is consistent with the result in Table 1, which suggests that HSV2 is probably the type present in Chagyrskaya 7. Our result in Table 2, with more reads mapped to the HSV1 reference sequence, is explained by the fact that the reads used in this analysis were from the assembly in Figure 1 and, therefore, were enriched with HSV1 reads. It must be noted, however, that the objective of the analysis presented in Table 2 was to show that reads that mapped to the assembly reference human alphaherpesvirus 1 (MN136523), which was used in Figure 1, also mapped consistently to the taxonomic references (NCBI RefSeq) of HSV1 and HSV2.

An analysis of deamination in the terminal ends of reads assembled to reference sequences showed that the expected pattern of aDNA could be observed (Figure 6). This is central to ruling out contamination with modern extant viral DNA. Furthermore, the sequence differences observed between inferred HAdV-7-N1, HSV1-N1 and HPV12-N1 suggest that they are distant from any extant viral genomes of the respective groups (Figure 7). Neighbor-joining trees obtained as results of blast analysis (Appendix A) were compared with topologies inferred by maximum likelihood (Appendix A) and Bayesian methods (Figure 7) detailing the relationships of the Neanderthal viruses with their respective ten closest extant relatives. The inferred Neanderthal sequences were separated in 100% bootstrap pseudoreplicates using maximum likelihood trees (Appendix A). Bayesian trees also supported the topologies and distances of inferred Neanderthal sequences and their extant relatives (Figure 7).

Although the assembly of adenovirus and papillomavirus showed that C-to-T deamination was significantly more abundant in the 5′ overhangs as compared to the other changes away from the termini, it can be argued that the C-to-T substitution rate of Neanderthal reads was around 30% higher than that in the results described by Skov et al. [22]. The estimates of aDNA damage were calculated (Figure 6) using mapDamage 2.0 [26], which implements a fast approximation of the DNA damage via a Bayesian framework and extends the Briggs–Johnson model of DNA damage [48]. The mapDamage model employs maximum likelihood to estimate DNA damage parameters. In the paper describing mapDamage 2.0 [26], the authors compare the two methods (Appendix A), showing that the estimates of cytosine deamination rates obtained by the mapDamage 2.0 program were consistently higher than those obtained using the Briggs–Johnson model. Since the estimates of Skov et al. [22] were based on Briggs et al. [48], it is expected that a difference in the estimates of the parameters based on these two different methods will be observed.

As discussed by Briggs et al. [48], when strand-equivalent reciprocal nucleotide substitutions are analyzed in DNA sequences from Pleistocene organisms, C-to-T changes are more frequent than G-to-A changes. Also, the rates of G-to-A and C-to T changes are higher than the rates of the other two transitions in contrast to DNA sequences determined from modern DNA. Deamination of cytosine (C) residues to uracil (U) residues in ancient DNA is responsible for the excess of C-to-T substitutions. The G-to-A could be caused by the deamination of guanine residues to xanthine (X) residues, which are read by methods of DNA sequencing as adenine residues (A), therefore causing the observed G-to-A misincorporations in ancient DNA.

It must also be noted that, in our analysis, we show that C-to-T changes are higher than G-to-A changes, and both are much higher in terminal positions than central positions (Figure 6). Also, in adenovirus and papilloma viruses, the C-to-T and G-to-A are less frequent in the internal sections as compared to terminal positions, which are fully consistent with ancient DNA patterns (Figure 6a,c). In the herpesvirus, the C-to-T and G-to-A are slightly higher than other changes in internal sections and significantly higher in terminal positions (Figure 6b).

The comparison of Neanderthal viral C-to-T and G-to-A in 5′-termini and 3′-termini with negative control (Figure 6a–d) showed that aDNA patterns were significantly different, as the C-to-T and G-to-A transitions in termini were 10× higher. The negative control consisted of 6,000,000 reads of a present-day Adenovirus-infected and sequenced cell culture. The comparison of the patterns of Neanderthal viruses C-to-T and G-to-A with the positive control showed that the HAdV-7-N1, HSV1-N1 and HPV12-N1 reads had about half of C-to-T at termini (5′ and 3′), but about five times more G-to-A. The positive control consisted of Neanderthal reads (from Chagyrskaya 7, the same pool used for viruses) mapped to the human reference mtDNA (Figure 7e). Although all viral patterns and mtDNA patterns were consistent with aDNA material, the base frequencies in the different references might explain the difference, since in mtDNA, G + C = 44.4 (C = 31.3%, G = 13.1%), and therefore, more C-to-T than G-to-A is expected in the termini of mtDNA reads because the sheer number of Cs as compared to Gs is more than double. In adenovirus, G + C = 51.2% (C = 25.9%, G = 25.3%); in herpesvirus (HSV1), G + C = 68.3% (C = 33.8%, G = 25.3%); in HSV2 G + C = 70.4% (C = 35%, G35.3%); and in papillomavirus, G + C = 42.4 (C = 20%, G = 22.4%).

The possibility that simple random similarities between Neanderthal reads and viral reference sequences could explain the mapping results shown in Figure 1 was tested with random mock reference sequences of adenovirus, herpesvirus and papillomavirus. The size and base composition of random references were identical to the actual counterparts. Mapping random references would indicate random noise in the analysis (Table 3). As shown, the assemblies obtained with real sequences had more reads mapped and higher coverage than the random sequences for all three viruses, as assessed by Welch’s *t*-test (Table 3). For adenovirus, the real reference had a coverage of 102.2 (180,419 reads mapped) as opposed to a coverage of 62.2 (126,613 reads mapped) for the random reference. For herpesvirus, the real reference had a coverage of 171.2 (1,224,613) as compared to a coverage of 154.8 (1,166,326 reads mapped). For papillomavirus, the real reference had a coverage of 115.4 (23,998 reads mapped) as opposed to the random reference coverage of 51.0 (22,682 reads mapped). All *p*-values of these comparisons rejected the null hypothesis (that the samples are equal), with *p*-values = 0 (Table 3). Also, in assemblies of random references, the numbers of regions with single-strand coverage were higher than with real references (Figure 1 and Figure 9). This indicates that random similarities between Neanderthal reads and viral sequences are substantial, and that the *bona fide* signal obtained via mapping of Neanderthal reads and actual viral genomes was above the level expected by random noise.

A negative control for nonpersistent DNA virus was included (Figure 12). This analysis revealed the relevance of using random reference sequences for assessing the level of random noise in this type of analysis. Mapping of Neanderthal reads to human parvovirus B19 reference was within the expected range for random similarities of short reads, as shown by Welch’s *t*-test (Table 4). Also, this indicates that not all DNA viruses were picked by our approach, but it is likely that our results reveal bona fide DNA viruses that produced persistent infection in the Neanderthal individual tested. The impact of this finding on populations, as suggested by Wolff and Greenwood [6], remains to be tested in future studies.

Additional tests for random similarities and short read lengths were performed, such as mapping to the human genome (hg38) (Figure 10) and reassembly to random reference sequences (Figure 11). Mapping of Neanderthal viral reads of assemblies in Figure 9 showed that they did not map to any human nuclear chromosomes, and only a small number of reads mapped to mtDNA (Figure 10). Regarding mapping to random reference sequences, we showed that when only the subset of reads that map to viral references was considered in the analysis, very few reads mapped to random references, which strongly suggests a particularly good signal-to-noise ratio in our analysis. Taken together, these tests (Figure 10 and Figure 11) further suggest that our results were not caused by random mapping or short read lengths. The difference between mapping to random reference sequences observed in Figure 9 and Figure 11 is explained by the high abundance of non-human DNA in the Neanderthal SRA raw data (Appendix A). The taxonomic analysis of sequence reads, based on MinHash-based k-mers implemented in SRA STAT, revealed that more than 90% of the reads in runs of Chagyrskaya 7 were not human DNA. In several run files, only between 0.5% to 1% were human DNA. Viruses comprised no more than 0.01%, and more than 90% were unidentified sequences comprising the dark matter (unidentified spots) shown in SRA Krona view (Appendix A) [49]. The analysis in Figure 9 includes “dark matter” reads, while in the analysis in Figure 11, these reads are filtered out.

The assessment of phylogenetic relationships of Neanderthal viral sequences to modern human counterparts and viral sequences of non-human primates showed that the inferred Neanderthal sequences followed the expected phylogenetic pattern, and no further analysis is necessary within the scope of this study. Therefore, as expected, and regarding RefSeq non-human viral sequences, when modern humans and Neanderthal viral sequences are analyzed, it is evident that chimpanzee’s viral sequences, and those of other primates, function as an outgroup to the Neanderthal and modern human ingroup (Appendix A). Murid sequences are a more distant outgroup relative to the primate ingroup. This is the expected pattern. We propose that a deeper, more detailed phylogenetic analysis in a future study would require the sequencing of viral genomes directly amplified from Neanderthal DNA to close the low-coverage gaps and confirm synonymous and non-synonymous SNPs to determine the intraspecific and interspecific relationships more precisely between these viral sequences.

Regarding the weight of deamination patterns in non-synonymous SNPs detected in Neanderthal viral genes, the effect of deamination was minimal. Most C-to-T and G-to-A changes were below the threshold for consensus sequence generation. In other words, the C-to-T and G-to-A changes were located preferentially at the termini of individual reads in a quantity and pattern compatible with aDNA, as shown above, but not all reads showed these changes (Appendix A). Although a deamination pattern was present, it was below 50% frequency. Therefore, the deamination bias did not have to be compensated for in phylogenetic analysis and did not compromise the inference of amino acid changes in viral DNA sequences. Deamination in ancient DNA was consistently around 15–20% of C-to-T changes in the termini of reads [8]. In the case of HSV1 retrieved from human remains dating back to 500 and 1500 years old, the levels of C-to-T deamination in 5′ termini were 20% and 12.5%, respectively. Although C-to-T deamination in 5′ termini was higher than in the middle of the reads, it was not 100%, and therefore, the C-to-T deamination can be differentiated from a bona fide C-to-T transition. Thus, it does not affect the inference of synonymous and non-synonymous changes in Neanderthal viral sequences.

A tentative classification of the Neanderthal viruses, inferred from our data, suggests that HAdV-7-N1 can be classified within the family *Adenoviridae*, the genus *Mastadenovirus* and the species of human Adv-B or Adv-C. The guidelines of the International Committee of Taxonomy of Viruses (ICTV Report, 2022) define that “if virus neutralization data are available, lack of cross-neutralization combined with a phylogenetic distance of more than 15% separates two types into different species. If the phylogenetic distance is less than 10%, any additional common grouping criteria from the list above may classify separate types into the same species even if they were isolated from different hosts” [50]. Therefore, based on these guidelines, there is no justification to propose that HAdV-7-N1 constitutes a new species. Neutralization data are not available, and the phylogenetic distance is not superior to 15% at either the nucleotide or amino acid levels in any regions of the genome. The distances between HAdV-7-N1 and any of its extant relatives is 5.1% on average (Appendix A). In CDS E2B (DNA Polymerase), the identity between HAdV-7-N1 and human adenovirus 7 (HAdV-7) is 95.7% (4.3% distance), and the ICTV species demarcation criteria in this genus depend on the phylogenetic distance (>10–15%, based primarily on distance matrix analysis of the DNA polymerase amino acid sequence) and host range. Therefore, based on both criteria, HAdV-7-N1 should be maintained in the same species of the genus *Mastadenovirus*, since the distances are within the 10% limit and they infect the same host. Although modern humans and Neanderthals constitute two different subspecies, they belong in the same species, namely, *Homo sapiens sapiens* and *Homo sapiens neanderthalensis*.

Nucleotide identity between HSV1-N1 and HSV1 and HSV2 does not suggest that HSV1-N1 is a new species, as shown in Figure 13. According to ICTV guidelines, the species demarcation, at present, uses sequence data flexibly to support taxonomic proposals based on phylogenetic grouping of viruses, does not require the availability of a complete genome sequence, does not depend on any specified gene or group of genes, and does not specify genetic distance thresholds for taxon differentiation [51].

In the case of HPV12-N1, the nucleotide distance to human papillomavirus strains RTRX7 and mSK144 was 18% at the whole-genome level. The HPV12-N1 seems to be a bona fide member of *Betapapillomavirus 1* and not a new species. According to ICTV, the species demarcation criterion, for a putative novel papillomavirus genome with complete genome sequence data available is that it is <70% related to papillomaviruses within the genus *Betapapillomavirus*, which was not observed in our data, where whole-genome nucleotide identities range from 74 to 82% within the group (Figure 7) [52].

The refinement of the classification of these three recovered Neanderthal viruses ultimately depends on the possibility of amplifying their genome segments via PCR (using primers based on highly conserved regions) directly from DNA extracted from Neanderthal bone material. Sequencing of these amplicons and incorporation of these reads into our assemblies can resolve nucleotide ambiguities and improve the sequencing accuracy of regions with low coverage.

Regarding the probability of finding all three viruses in the same sample of a single individual, it has been shown that single individuals of present-day humans are exposed to about 10 viral species over their lifetimes [53]. Particularly in the case of lifelong infections, it is not surprising to verify that an individual has been exposed to these three viruses, because they are highly contagious and associated with social behavior [6]. As a matter of fact, proviral remnants of ancient retroviral infection of germ cells now make up about 8% of the human genome in the form of human endogenous retroviruses (HERVs), which indicates that exposition of humans to viral agents is extensive [54]. A single sample of one individual might contain remnants of lifelong infections, and this probability depends on many external factors associated with the distribution of these viruses in tissues, as well as environmental factors affecting the preservation of this material. The assessment of individual coinfection probability with these three different viruses, based on the data at hand, is highly speculative and depends on the specific epidemiological patterns of the Chagyrskaya population, a conjecture that is beyond the scope of this study.

Variable regions identified in these reconstructed viral genomes could reveal adaptive signatures of Neanderthals, such as nonsynonymous substitutions in viral surface proteins that interact with host cell receptors. Conserved regions can be exploited as potential PCR targets for amplification of these viral genomes from isolated Neanderthal DNA.

## 5. Conclusions

In this study we demonstrate, as a proof of concept, the possibility to detect vestigial viral DNA from sequence reads of Neanderthal genome projects. This would be the first step in addressing the hypothesis that DNA viruses that produce persistent infections might have had an impact on the extinction of Neanderthals. We show that remnants of adenovirus, herpesvirus and papillomavirus could be detected in Neanderthal genome data, above the level of pure random similarities, given the read lengths and assembly coverages considered herein. Under the same conditions, a nonpersistent DNA virus was not detected above the level of pure random similarity. These results could be taken in the context of the hypothesis by Wolff and Greenwood [6] that adenovirus, herpesvirus and papillomavirus infected Neanderthals, which, in turn, could have contributed to processes associated with the Neanderthals’ extinction.

## Figures and Tables

**Figure 1 viruses-16-00856-f001:**
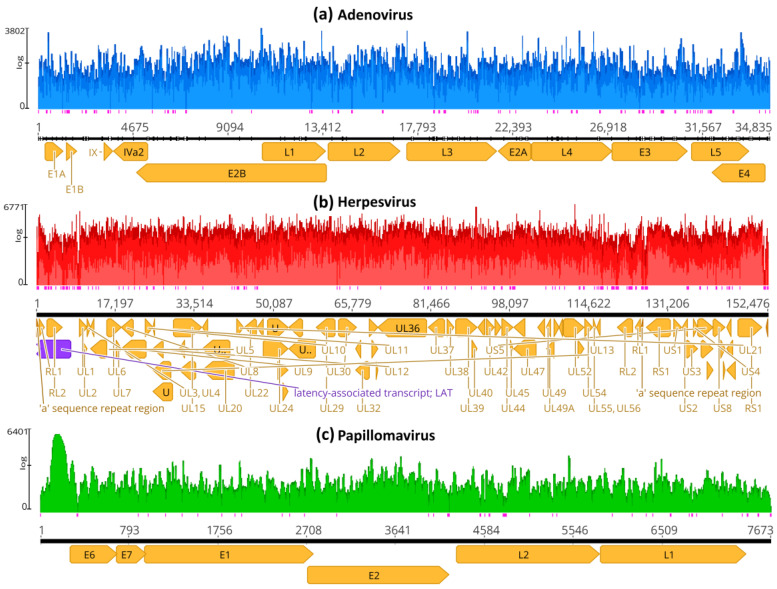
Mapping coverage of herpesvirus, papillomavirus and adenovirus using Neanderthal sample-derived reads. Reference mapping was performed using BBMap [24,25]. (**a**) Mapping of 180,419 Neanderthal reads (mean coverage 102.2) to adenovirus assembly reference (or template) (KX897164). (**b**) Mapping of 1,224,413 reads (mean coverage 171.2) to herpesvirus assembly reference (MN136523). (**c**) Mapping of 23,998 Neanderthal reads (mean coverage 115.4) to papillomavirus assembly reference (X74466). Pink vertical bars below the assemblies indicate single-strand regions, and orange arrows indicate the protein-coding genes. The mapping was performed with the same read dataset and these three reference sequences (templates) simultaneously, as detailed in Materials and Methods, but each depiction shows the corresponding assembly reference sequence of each bam file with the corresponding mapped reads.

**Figure 2 viruses-16-00856-f002:**
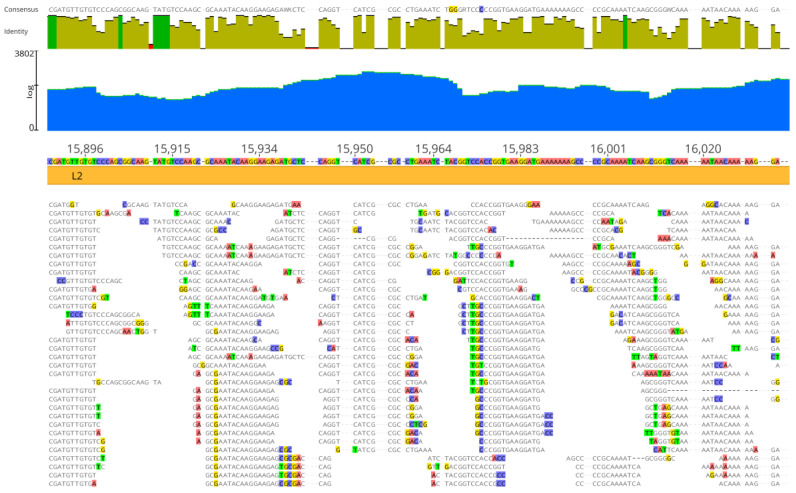
Details of Neanderthal reads (Chagyrskaya 7) mapped to adenovirus assembly reference genome sequence (GenBank KX897164). Blue area indicates coverage, dark green vertical bars indicate 100% similarity, olive green vertical bars indicate nucleotide identity <100% and ≥30% and red vertical bars indicate <30% identity. Colored residues are those that differ from the assembly reference. The consensus (50% strict rule) is above the nucleotide identity bars. The mapping was performed with three reference sequences simultaneously, as detailed in Materials and Methods, but the depiction is shown with the adenovirus assembly reference sequence for clarity.

**Figure 3 viruses-16-00856-f003:**
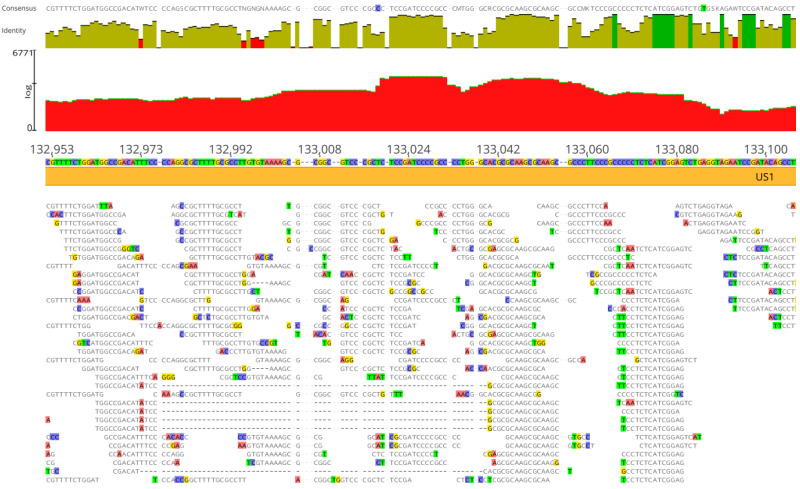
Details of Neanderthal reads (Chagyrskaya 7) mapped to herpesvirus assembly reference genome sequence (GenBank MN136523). Red area indicates coverage, dark green vertical bars indicate 100% similarity, olive green vertical bars indicate nucleotide identity <100% and ≥30% and red vertical bars indicate <30% identity. Colored residues are those that differ from the assembly reference. The consensus (50% strict rule) is above the nucleotide identity bars. The mapping was performed with three reference sequences simultaneously, as detailed in Materials and Methods, but the depiction is shown with the herpesvirus assembly reference sequence for clarity.

**Figure 4 viruses-16-00856-f004:**
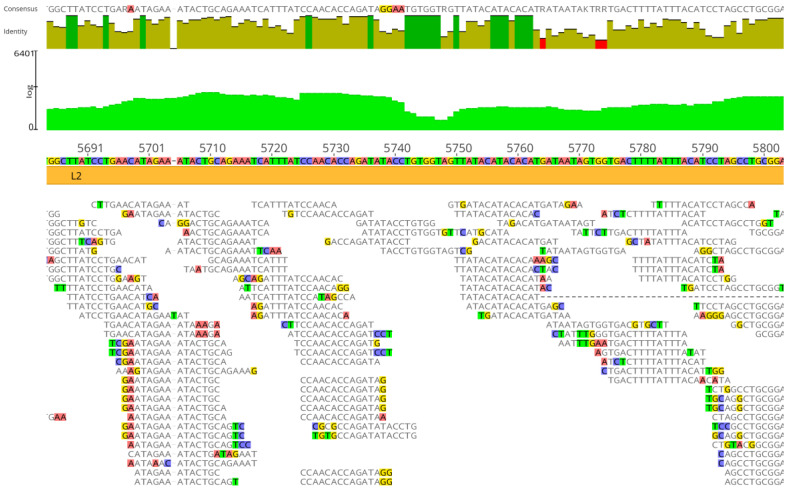
Details of Neanderthal reads (Chagyrskaya 7) mapped to papillomavirus assembly reference genome sequence (GenBank X74466). Light green area indicates coverage, dark green vertical bars indicate 100% similarity, olive green vertical bars indicate nucleotide identity <100% and ≥30% and red vertical bars indicate <30% identity. Colored residues are those that differ from the assembly reference. The consensus (50% strict rule) is above the nucleotide identity bars. The mapping was performed with three reference sequences simultaneously, as detailed in Materials and Methods, but the depiction is shown with the papillomavirus assembly reference sequence for clarity.

**Figure 5 viruses-16-00856-f005:**
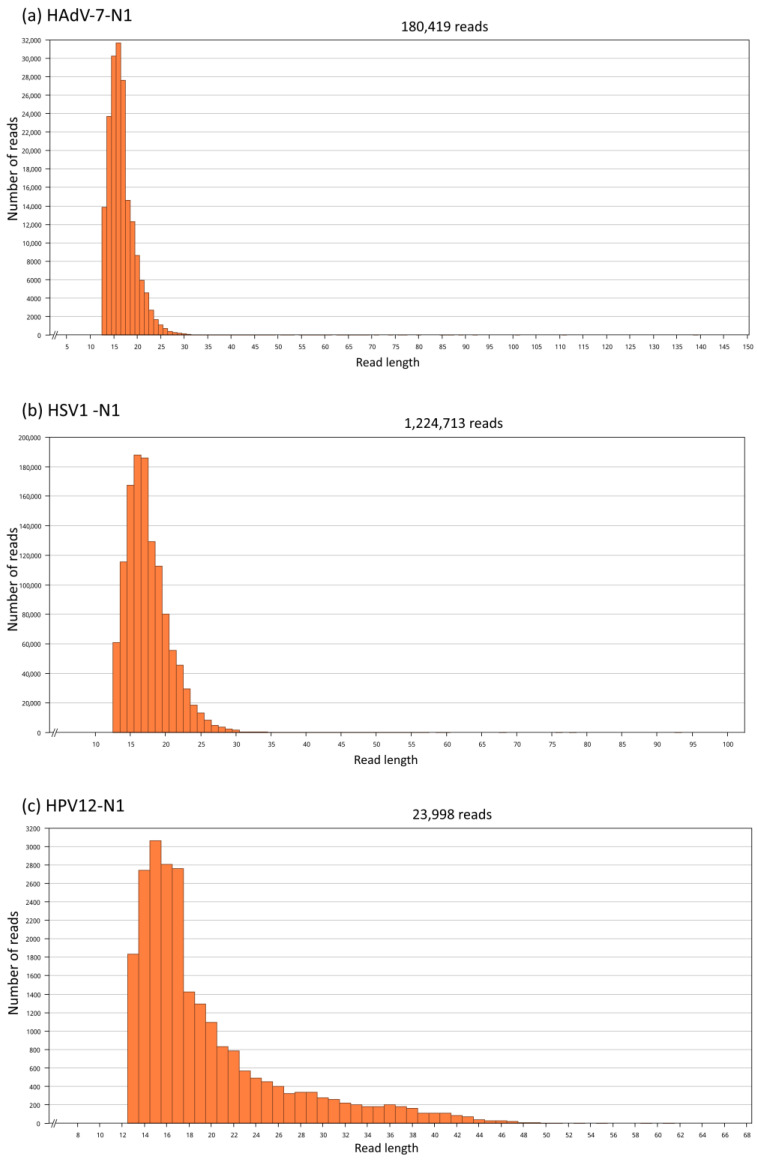
Neanderthal read length distributions. Read lengths used for mapping of (**a**) HAdV-7-N1 (from 13 to 139 bases), (**b**) HSV1-N1 (from 13 to 93 bases) and (**c**) HPV12-N1 (from 13 to 61 bases).

**Figure 6 viruses-16-00856-f006:**
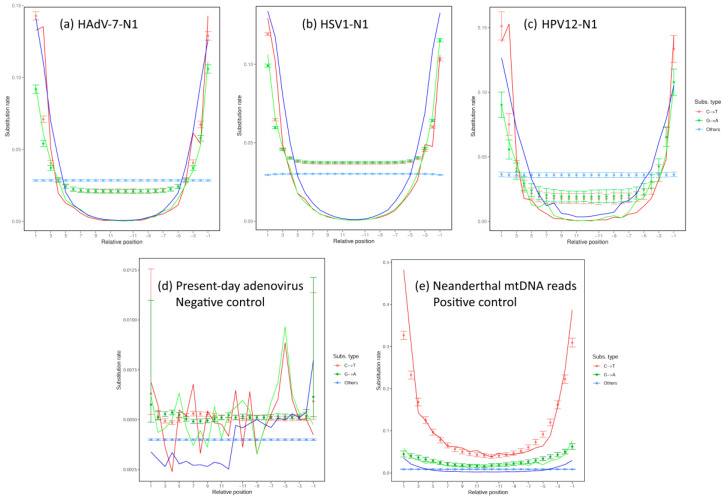
Substitution types and their relative positions in reads used for mapping of the HAdV-7-N1 assembly (**a**), the HSV1-N1 assembly (**b**) and the HPV12-N1 assembly (**c**). The BAM files of the assemblies were analyzed using the mapDamage program [26]. Discrete points per relative position indicate the substitution rate per read position of Chagyrskaya 7, and lines indicate the likelihood curves. In all three assemblies, the C-to-T deamination occurred at the terminal overhangs, as expected in ancient DNA reads. G-to-A changes were produced in ancient DNA by deamination of G to xanthine (X) which was read by sequencing methods as A. The number of reads analyzed for HAdV-7-N1 was 180,419; for herpesvirus, 1,224,413 reads; and for papillomavirus, 23,998 reads. The abscissa indicates the substitution type relative to the termini of reads, 1 being the 5′ termini and up to 11 positions inwards, while the negative numbers provide the opposite end. A negative control (**d**) consisted of sequence reads from a present-day sample of adenovirus (SRA run SRR26630751) mapped to human adenovirus 7 reference (AC_000018.1) (6,205,335 mapped reads). The positive control (**e**) consisted of Neanderthal reads of Chagyrskaya 7 mapped to the human mitochondrial genome reference sequence rCRS (NC_012920.1) (29,500 mapped reads). For negative and positive controls, the BBMap (mapping) and BBDuk (trimming) parameters used were identical, with HAdV-7-N1, HSV1-N1 and HPV12-N1.

**Figure 7 viruses-16-00856-f007:**
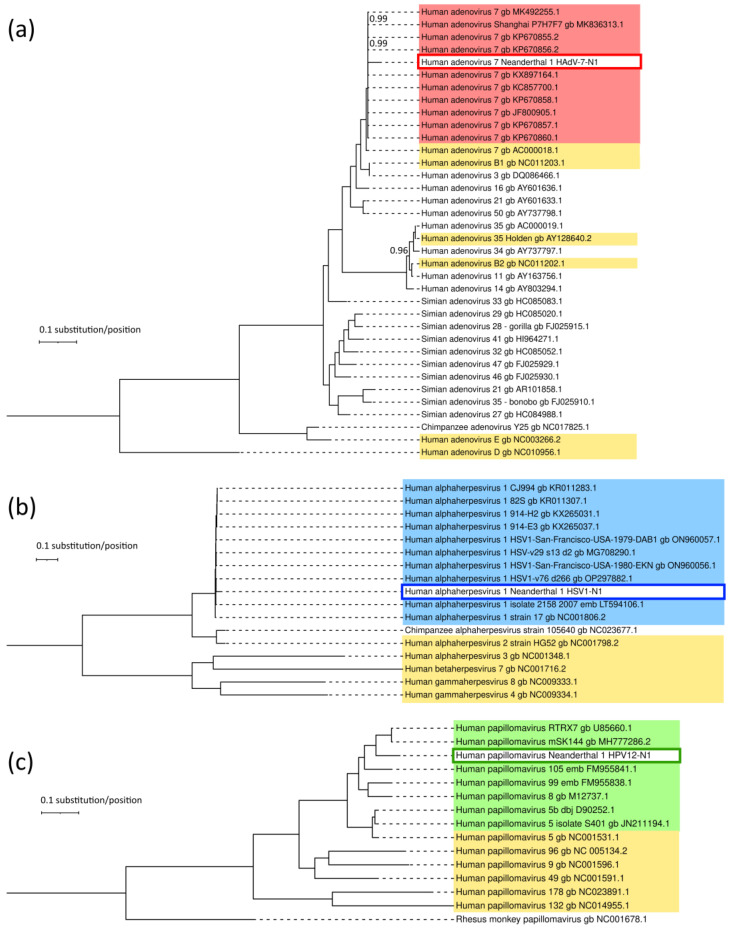
Bayesian trees of Neanderthal viral genomes and extant relatives. The genealogies of (**a**) HAdV-7-N1, its closest relatives, human RefSeq, Simian AdV and main HAdV serotypes. (**b**) HSV1-N1, its closest relatives and human RefSeq. (**c**) HPV12-N1, its closest relatives and human RefSeq. All posterior probabilities of branch clusters are 1 unless otherwise indicated. Scale bars indicate substitutions per position. The likelihoods of trees are (**a**) LnL= −249,040.728, (**b**) LnL= −1,187,183.249 and (**c**) LnL= −76,429.536. Terminal taxa are identified with accession numbers. Whole genomes were aligned using MAFFT and used as input for Bayesian tree inference with the MrBayes program version 3.2.7. Red, blue and green boxes indicate the subtrees with sequences of the closest extant relatives of HAdV-7-N1, HSV1-N1 and HPV12-N1, respectively, as identified via MegaBlast analysis, and yellow boxes indicate the human NCBI RefSeq sequences for the corresponding viral taxa.

**Figure 8 viruses-16-00856-f008:**
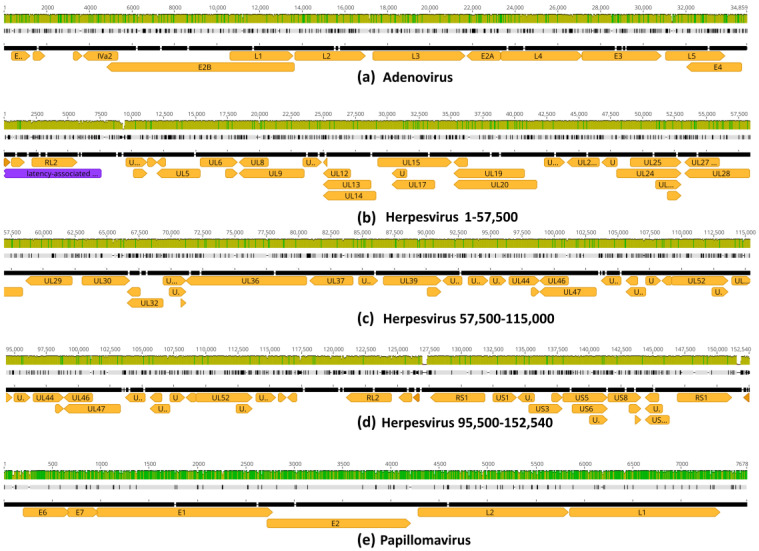
Alignment of reconstructed Neanderthal viral genomes to extant relatives. (**a**) HAdV-7-N1 vs. adenovirus (HAdV-7 KX897164) with 94.4% global nucleotide identity. (**b**–**d**) HSV1-N1 vs. alphaherpesvirus 1 (HSV1 MN136523) with 94.1% nucleotide identity. (**e**) HPV12-N1 vs. papillomavirus (HPV12 X74466) with 94.5% nucleotide identity. Pairwise alignments were performed using the Geneious aligner. Olive green bars indicate partial nucleotide identity while dark green bars indicate 100% nucleotide identity. Genes are indicated by orange arrows. Solid black lines are reference sequences of extant genomes. Black tick marks on grey lines indicate sequence differences of Neanderthal viruses relative to reference sequences.

**Figure 9 viruses-16-00856-f009:**
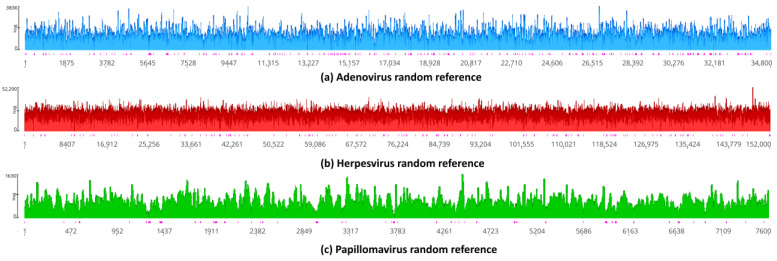
Assembly by mapping of Neanderthal reads against random reference sequences. Random reference sequences of adenovirus, herpesvirus and papillomavirus were obtained using UGENE version 44 [29]. Random sequences had the same sizes and base frequencies as actual reference sequences. Random reference sequences were used for genome assembly by mapping using the same programs and parameters as used for actual reference sequences. Pink vertical bars below coverage indicate single-strand regions.

**Figure 10 viruses-16-00856-f010:**
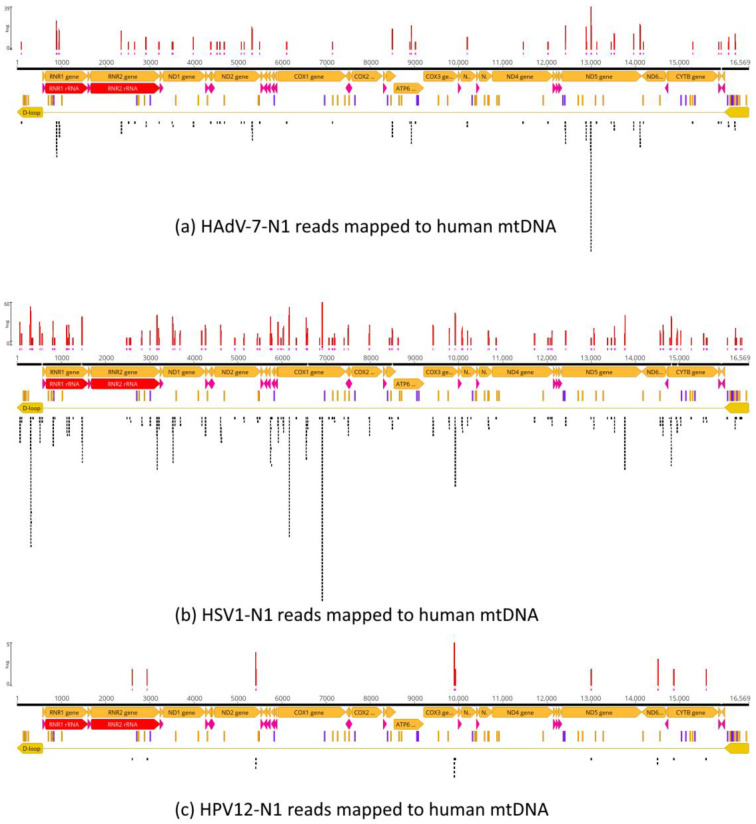
Mapping of HAdV-7-N1 (**a**), HSV1-N1 (**b**) and HPV12-N1 (**c**) reads against the human genome (hg38) reference sequence. Reads that mapped to references in Figure 1 were “unmapped” and used for mapping the twenty-five chromosomes of the human genome (22 autosomes, X, Y and mtDNA). No reads mapped to autosomes. Only mtDNA hits are depicted. Genome assembly by mapping used the same programs and parameters as those used for actual reference sequences. Pink vertical bars below coverage indicate single-strand regions.

**Figure 11 viruses-16-00856-f011:**
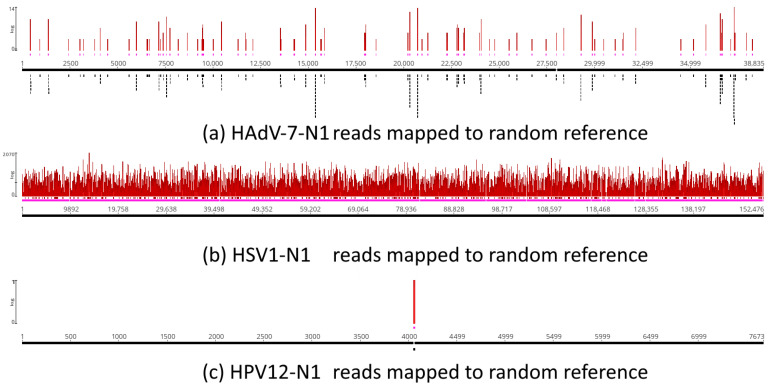
Mapping of HAdV-7-N1 (**a**), HSV1-N1 (**b**) and HPV12-N1 (**c**) reads against equivalent random reference sequences. Reads that mapped to references in Figure 1 were “unmapped” and used for mapping random reference sequences. Random sequences had the same sizes and base frequencies as actual reference sequences. Random reference sequences were used for genome assembly by mapping using the same programs and parameters as those used for actual reference sequences. Pink lines below coverage indicate single-strand regions.

**Figure 12 viruses-16-00856-f012:**
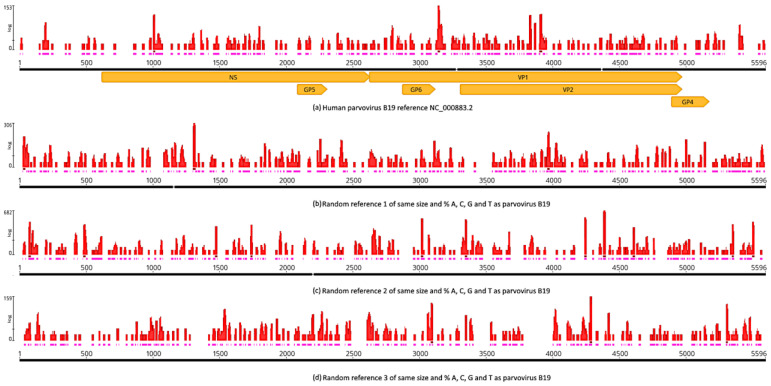
Mapping of Neanderthal reads to human parvovirus B19, a DNA virus that produces nonpersistent infection. (**a**) Coverage and read distribution along the parvovirus reference sequence. (**b**–**d**) Coverage and read distribution along random reference sequences of the same size and base frequencies as the parvovirus B19 reference sequence NC_000883.2 (NCBI RefSeq). Red bars above reference sequences indicate coverage, pink bars just below red bars indicate single strands and orange arrows indicate the parvovirus genes. Mapping was obtained with BBMap using the same parameters used herein for adenovirus, herpesvirus and papillomavirus.

**Figure 13 viruses-16-00856-f013:**
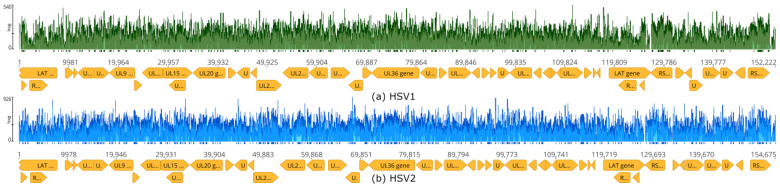
Mapping of Neanderthal reads to HSV1 and HSV2 reference sequences. The total set of 1.2 billion reads of Neanderthal Chagyrskaya 7 was assembled simultaneously to HSV1 NC_001806.2 (**a**) and HSV2 NC_001798.2 (**b**) reference sequences using BBMap. Green vertical bars indicate HSV1 coverage; blue vertical bars indicate HSV2 coverage. Orange arrows indicate genes. Tick marks below coverage bars indicate coverage above 50×. A total of 1,221,442,735 Neanderthal Chagyrskaya 7 reads were assembled to reference sequences to produce an HSV1 contig with 74,405 reads and an HSV2 contig with 105,046 reads. A total of 1,221,263,284 reads were not assembled.

**Table 1 viruses-16-00856-t001:** Viral segments identified by Sequence Taxonomy Analysis Tool (STAT) in Neanderthal Chagyrskaya 1 (SRA BioSample SAMEA110640442) and 7 (SRA BioSample SAMEA110640445) in SRA BioProject PRJEB55327. Includes only human pathogenic DNA viruses producing persistent infections. Excludes bacteriophages, environmental viruses, retroviruses, human endogenous retroviruses and RNA viruses. Removal of duplicate reads, primers and adapters resulted in the exclusion of approximately 50% of the reads. Ten representative samples of Chagyrskaya 7 are shown in the table. Viral reads comprise approximately 0.01% of each run.

Run	Chagyrskaya	# Reads		DNA Viruses
ERR10073004	01	5,187,065		human gammaherpesvirus 4, human papillomavirus 12 and 36
ERR10073005	01	1,106,287		human papillomavirus 20
ERR10073006	01	12,197,881		human gammaherpesvirus 4, hepatitis B virus, human papillomavirus 10, 12, 19 and 209
ERR10073007	01	5,384,224		*Gammapapillomavirus* 12
ERR10073008	01	581,197		Not detected
ERR10073009	01	5,434,074		human alphaherpesvirus 1
ERR10073010	01	11,535,641		human gammaherpesvirus 4, human papillomavirus 36 and human adenovirus 7
ERR10073011	01	2,145,096		Not detected
ERR10073012	01	813,154		Not detected
ERR10073013	01	5,836,042		*Papillomaviridae*
ERR10073014	01	10,283,890		human papillomavirus 12
ERR10073060	07	31,299,145		human papillomavirus 18, human respirovirus 1 and human mastadenovirus C
ERR10073061	07	30,544,456		human mastadenovirus C and human papillomavirus 16
ERR10073062	07	31,517,112		human mastadenovirus C
ERR10073063	07	34,004,864		human alphaherpesvirus 2, *Betaherpesvirinae* Stealth virus 1 and human mastadenovirus C
ERR10073064	07	34,089,383		human mastadenovirus C and human papillomavirus 16
ERR10073065	07	34,548,780		human betaherpesvirus 7 and *Cytomegalovirus* 1, *Mastadenovirus*
ERR10073066	07	34,530,209		human mastadenovirus C and human papillomavirus 16
ERR10073067	07	34,336,757		human alphaherpesvirus 2 and *Mastadenovirus*
ERR10073068	07	34,800,717		*Mastadenovirus*
ERR10073069	07	35,065,248		human mastadenovirus C

**Table 2 viruses-16-00856-t002:** Assembly of Neanderthal reads to NCBI viral reference sequences (NCBI RefSeq). A total of 14 adenovirus, 6 herpesvirus and 45 papillomavirus NCBI RefSeq (https://www.ncbi.nlm.nih.gov/refseq/ accessed on 23 January 2024) sequences were used for assembly by mapping with Neanderthal reads using BBMap, as described in Material and Methods. Three assemblies were generated, one for each viral family, and in each assembly, all sequences of each family were mapped simultaneously to all references in that family. The top 6 results, ranked by the number of mapped reads, are shown.

RefSeq Accession	# Mapped Reads		GenBank ID	Virus
Adenovirus				
AC_000018.1	78,255		GI:56160876	human adenovirus 7
NC_011203.1	54,835		GI:197944726	human adenovirus B1
NC_011202.1	8854		GI:197944766	human adenovirus B2
AC_000019.1	8573		GI:56160914	human adenovirus 35
NC_003266.2	6449		GI:51527264	human adenovirus E
NC_010956.1	2716		GI:190340974	human adenovirus D
Herpesvirus				
NC_001806.2	937,258		GI:820945227	human herpesvirus 1 17
NC_001798.2	167,080		GI:820945149	human herpesvirus 2 HG52
NC_009334.1	13,928		GI:139424470	human herpesvirus 4
NC_009333.1	10,443		GI:139472801	human herpesvirus 8 GK18
NC_001348.1	4056		GI:9625875	human herpesvirus 3
NC_001716.2	1857		GI:51874225	human herpesvirus 7
Papillomavirus				
NC_001531.1	1889		GI:9627145	human papillomavirus 5
NC_001591.1	494		GI:9627363	human papillomavirus 49
NC_005134.2	167		GI:50253426	human papillomavirus 96
NC_023891.1	82		GI:607064610	human papillomavirus 178
NC_001596.1	77		GI:9627396	human papillomavirus 9
NC_014955.1	76		GI:319962668	human papillomavirus 132

**Table 3 viruses-16-00856-t003:** Comparison of mapping with real assembly references, random references and Neanderthal sequence reads. Real HAdV-7-N1, HSV1-N1 and HPV12-N1 templates used were the same used in the assemblies in Figure 1: Human adenovirus 7 (KX897164), human alphaherpesvirus 1 (MN136523) and human papillomavirus type 12 (X74466). SD = standard deviation and # reads = mapped reads. Random reference sequences used here are shown in Appendix A. The column “Welch’s *t*” indicates the *t* statistics of Welch’s *t*-test. *c.v.* = critical value for infinite degrees of freedom and *p*-value is the probability that the null hypothesis (that both samples are statistically equivalent) is correct.

	Real			Random					
Coverage	Mean	SD	# Reads	Mean	SD	# Reads	Welch’s *t*	*c.v.*	*p*-Value
HAdV-7-N1	102.2	249.4	180,419	62.2	180.2	126,613	51.58	1.64	0
HSV1-N1	171.2	247.4	1,224,713	154.8	704.3	1,166,326	23.78	1.64	0
HPV12-N1	115.4	609.1	23,998	51.0	116.0	22,682	16.07	1.64	0

**Table 4 viruses-16-00856-t004:** Comparison of mapping Neanderthal sequence reads with human parvovirus B19 (NC_000883.2) reference and random references. SD = standard deviation and # reads = mapped reads. Random reference sequences used herein are shown in Appendix A. The column “Welch’s *t*” indicates the *t* statistics of Welch’s *t*-test. *c.v.* = critical value for infinite degrees of freedom. *p*-value (one-tailed) is the probability of rejecting the correct null hypothesis.

Reference	Coverage Mean	Coverage SD	# Reads	Welch’s *t*	*c.v.*	*p*-Value
Parvovirus B19	2.0	9.2	714	-	-	-
Random Appendix A	3.5	16.9	1271	−2.56	1.64	0.005267
Random Appendix A	7.2	41.6	2595	2.25	1.64	0.012046
Random Appendix A	2.7	10.0	975	−1.48	1.64	0.068387

## Data Availability

All data produced by analyses in this study are available upon request after acceptance of the manuscript for publication.

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
