# Peer review of "Reconstructing Prehistoric Viral Genomes from Neanderthal Sequencing Data"

_viruses, 2024, doi:10.3390/v16060856_

Round 1
Reviewer 1 Report (Previous Reviewer 1)
Comments and Suggestions for Authors
The manuscript by Ferreira et al is a resubmission of an extensively revised, and earlier rejected paper, which now shows major improvements. The authors have done additional analyses to classify the viral N-strains, and validate their results.
Overall, I have only a minor remark, and a suggestion:
The first remark concerns the abstract, in which it is stated that HPV is a dsDNA virus that establishes lifelong latency and causes persistent infections. However, this should be modified, as most HPV infections are cleared, even when this may take years. The wording in the introduction is much more cautious, so only those statements in the abstract need adaptation to explain that it ‘may establish’ and ‘may cause’ etc. Or, possibly 'often causes'
The second remark is on HBV infection, another potentially chronic virus infection, which is nicely mentioned in the introduction. HBV mostly establishes chronic infections when the virus was introduced during early childhood. When adults get infected, the infection is commonly cleared. I noted from Table 1 that HBV-related reads were detected in Chagyrskaya 1, but not in Chagyrskaya 7, the sample examined here in detail. Since HBV DNA has been successfully retrieved from several ancient samples, would testing for HBV reads not nicely illustrate the validity of your other virus results? When no HBV reads can indeed be detected in Chagyrskaya 7, it would increase the credibility of finding the other three viruses. On the other hand, even when the STAT analysis has been negative, it would not be unlikely to find HBV in ancient samples.
Comments on the Quality of English LanguageEnglish is fine as far as I can see with all the track changes. Final check can be done at the production stage
Author Response
Reviewer 1
The manuscript by Ferreira et al is a resubmission of an extensively revised, and earlier rejected paper, which now shows major improvements. The authors have done additional analyses to classify the viral N-strains, and validate their results.
R: We thank the reviewer for assessing the manuscript.
Overall, I have only a minor remark, and a suggestion:
The first remark concerns the abstract, in which it is stated that HPV is a dsDNA virus that establishes lifelong latency and causes persistent infections. However, this should be modified, as most HPV infections are cleared, even when this may take years. The wording in the introduction is much more cautious, so only those statements in the abstract need adaptation to explain that it ‘may establish’ and ‘may cause’ etc. Or, possibly 'often causes'
R: Corrected. We replaced “…double stranded DNA viruses that establish lifelong latency and can, produce persistent infections” for “…double stranded DNA viruses that may establish lifelong latency and can, produce persistent infections”.
The second remark is on HBV infection, another potentially chronic virus infection, which is nicely mentioned in the introduction. HBV mostly establishes chronic infections when the virus was introduced during early childhood. When adults get infected, the infection is commonly cleared. I noted from Table 1 that HBV-related reads were detected in Chagyrskaya 1, but not in Chagyrskaya 7, the sample examined here in detail. Since HBV DNA has been successfully retrieved from several ancient samples, would testing for HBV reads not nicely illustrate the validity of your other virus results? When no HBV reads can indeed be detected in Chagyrskaya 7, it would increase the credibility of finding the other three viruses. On the other hand, even when the STAT analysis has been negative, it would not be unlikely to find HBV in ancient samples.
R: We thank the reviewer for the suggestion, but we will not perform novel analyses in this study since it is already too extensive. This suggestion will be the basis for a future study for sure in which, if possible, we would include the reviewer as co-author. We believe we provided sufficient controls for the three viruses analyzed plus a negative control (parvovirus).
Comments on the Quality of English Language
English is fine as far as I can see with all the track changes. Final check can be done at the production stage.
R: We revised extensively the manuscript with the Microsoft Word spelling and grammar check.
Reviewer 2 Report (Previous Reviewer 2)
Comments and Suggestions for Authors
According to this research, it can be concluded that viral sequences corresponding to persistent DNA viruses (Herpesvirus, Adenovirus, Papillomavirus) can be found in ancient DAN sequencing data containing Neanderthal DNA sequences.
But far too much space in the introduction and conclusion chapters is devoted to hypotheses 1) That these viruses could have been introduced by modern humans among the Neanderthals. Actually, there are no evidences that these viruses were not already present in their common ancestor, or that it was Neanderthal populations who introduced these viruses into modern humans. 2) No argument supports the hypothesis that these viruses were one of the causes of the disappearance of Neanderthal populations. Actually It is not these viruses but influenza viruses which are one of the major causes of the terrible epidemics introduced by Europeans among Indian-Americans in the sixteenth century, etc.. Therefore, the discussion regarding these hypotheses must be significantly reduced and must stick mainly to the facts.
Comments on the Quality of English Language
No comments
Author Response
Reviewer 2
According to this research, it can be concluded that viral sequences corresponding to persistent DNA viruses (Herpesvirus, Adenovirus, Papillomavirus) can be found in ancient DAN sequencing data containing Neanderthal DNA sequences.
R: We thank the reviewer for taking the time to evaluate our study.
But far too much space in the introduction and conclusion chapters is devoted to hypotheses 1) That these viruses could have been introduced by modern humans among the Neanderthals. Actually, there are no evidences that these viruses were not already present in their common ancestor, or that it was Neanderthal populations who introduced these viruses into modern humans. 2) No argument supports the hypothesis that these viruses were one of the causes of the disappearance of Neanderthal populations. Actually It is not these viruses but influenza viruses which are one of the major causes of the terrible epidemics introduced by Europeans among Indian-Americans in the sixteenth century, etc.. Therefore, the discussion regarding these hypotheses must be significantly reduced and must stick mainly to the facts.
R: We agree with the reviewer but the hypothesis proposed by Wolff and Greenwood (doi.org/10.1016/j.mehy.2010.01.048) was the staring point of our search and therefore we had to give the readers a sufficient context to our search. The hypothesis was mentioned only once in the “Discussion”, once in “Conclusions” and once in the “Introduction”. We maintained the reference to this hypothesis in the “Introduction” since it was the starting point for our search and only once in the “Conclusions” with a more cautious rephrasing. We agree that it can be argued that no data supports the hypothesis that these viruses were one of the causes of the disappearance of Neanderthal populations and that influenza might play a role. However, since influenza does not produce persistent infections the likelihood of the presence of remnants of influenza in ancient DNA of this age would be unlikely. No influenza reads were detected in the metagenomic STAT analysis. We rewritten the introduction including the reviewers concerns.
Comments on the Quality of English Language
No comments
Reviewer 3 Report (New Reviewer)
Comments and Suggestions for Authors
Ferreira et al. produced an interesting work on „reconstructing prehistoric viral genomes from Neanderthal sequences”. They could identify “ancient” adeno-, herpes-, and papillomavirus genomes by data mining from Homo sapiens neanderthalensis samples. They took care to prove that these sequences are really from ancient viruses and not from present day viruses. While these genomes still contain gapes and ambiguities, they give adequate information about their closest extant relatives. Thus, the results are useful and very interesting. However, some improvement would raise the usefulness and correctness of the MS. First, the MS is way too long. The 34 pages contains many repetitions that should be deleted. E.g., in the Results section the methods should not be repeated, neither the results in the Discussion. The Supplementary information contains too much details (51 files, 177 Mb, one of the files is 146 Mb long). Scientists always have a (hard) task to filter out the most important data and final conclusions to present for the readers to keep his interest continuous.
A general problem is the misinterpretation of virus species concept, and thus mixing of the sequenced actual viruses with the species where they belong to. Virus species is an abstract idea about a group collecting the “adequately” similar virus types. The authors should name the virus types like human adenovirus 7 (never starting with capital) or “members of a species, e.g., that of Human mastadenovirus B [in italics]” Presently the MS applies a mixture of virus names and species names (for adenoviruses the real virus name ends with an Arabic number and the species with a capital letter). This means that they cannot “sequence a virus species”, while the human adenovirus 7 should not be written in italics (Fig. 7). (Only official virus taxon names are be be written in italics, but including also the family and higher viral taxon names, like Papillomaviridae. When you speak about a “herpesvirus”, it shouldn’t start with a capital!) The origin of this mixture of names may be that GenBank applies different names for the “organism”, sometimes the types, while other times the species. Still, a uniform labelling (applying the type numbers or strain designations) for the sequences is the correct practice.
Your final conclusion that N-adenovirus would not merit the establishment of a new species for it is absolutely correct, however, you should not cite the criteria of the obsolete 2011 ICTV book. The presently official ICTV taxonomy for AdVs (2022) requires a more than 10-15% difference in the polymerase amino acid sequence (not 5-15%) for establishing a new species.
On the phylogenetic trees, the authors should show the closest serotypes as they are the most relevant. N-adenovirus proved to be a HAdV-7 variant, thus all the other types of species Human mastadenovirus B should be shown on the tree to demonstrate their distinctness (after removing the ambiguities in your sequences): HAdV-3, -11, -14, -16, -21, -34, -35, -50, and including (at least some of) the simian (chimpanzee and gorilla AdV) members of HAdV-B. This is important as actually this lineage is presumed to be the real (continuously coevolved) gorilla lineage, just two of its ancient members switched to humans producing the B1 and the B2 “subgroups” (by now differentiated already to different serotypes) while other gorilla AdVs switched to chimpanzees. On the phylogenetic tree, you mislabelled NC010956 as B2; it is already another species namely Human mastadenovirus D. (Actually, HAdV-D is the real human lineage, which constantly coevolved with Homo sapiens). Unfortunately, you did not include any (or some/all members, i.e., at least 4 human, and about 10 chimpanzee, bonobo and gorilla AdV types) from Human mastadenovirus C in spite that sometimes you found HAdV-C members to be most similar to your Neanderthal sequences.
Some special notes:
Adenovirus does not establish real “latency”. Reword these sentences.
“(IV) Adenoviruses [11].” This reference is about the HAdV-C but not about the whole family.
„evolution of transmissible and nontransmissible diseases” Are there any other possibility?
„≈70% identical with chimpanzee sequences” Which chimpanzee AdV sequence? Y25? What about the simian AdV-21? It is also a chimpanzee AdV and furthermore it is exactly in the species Human mastadenovirus B. You should compare your N-adenovirus to the closest chimpanzee and gorilla AdV types.
„family Adenoviridae, genus Mastedenovirus and in species human Adv-B or Adv-C” should be „family Adenoviridae [in italics!], genus Mastadenovirus and in Human mastadenovirus B or Human mastadenovirus C” (or HAdV-B or HAdV-C /capital V for HAdV/!) (Theoretically, you should not abbreviate a species name, but it is done generally.)
„prototype new species”. There is no „prototype” species anymore: „candidate member for a new species”
„adenovirus, herpesvirus and papillomavirus infected Neanderthals” These viruses were present already in all reptiles and mammals thus it is not too probable that they would be a new infection for this human subspecies. Maximum „certain very pathogenic variants” could be new for Neanderthals.
Ref. 50. No authors/editors, nor Publisher for this obsolete old book (9th Report”)? Is it right to cite a whole book instead of its appropriate chapters? Is it correct to cite a bookstore item instead of the original scientific publication?
Fig. 7. The green background should be one line larger.
Comments on the Quality of English LanguageSome minore mistakes:
hypothesis .... "propose" should be "proposes"
"sequencing reads" should be rather "sequence reads"
"Human adenoviruses persistence" should be "Human adenovirus persistence"
"thresholds for taxa differentiation" > ... taxon ...
this "probabilitydependensd" on > ... depends on ...
"nucleotide similarity" > ... identity
"of theassemblies" > of the assemblies
"N-herpesvirus e N-papillomavirus" > ... and ...
Author Response
Response to reviewer #3
- Ferreira et al. produced an interesting work on „reconstructing prehistoric viral genomes from Neanderthal sequences”. They could identify “ancient” adeno-, herpes-, and papillomavirus genomes by data mining from Homo sapiens neanderthalensis samples. They took care to prove that these sequences are really from ancient viruses and not from present day viruses. While these genomes still contain gapes and ambiguities, they give adequate information about their closest extant relatives. Thus, the results are useful and very interesting. However, some improvement would raise the usefulness and correctness of the MS.
R: We are deeply thankful for the extensive and detailed analysis by reviewer # 3. All points raised are relevant and will certainly improve the quality of the study. We learned a lot from reviewers’ comments and tried to comply with all suggestions, particularly regarding details and formalism on the taxonomy of viruses. However, some of the comments and suggestions are conflicting with comments and suggestions by other reviewers and we sincerely tried to adequate the manuscript to comply with all reviewers. We cannot favour one reviewer over another and therefore the manuscript in its present edited form is a sum-up of various comments and requests made by scientists of different fields and expertise such as genomics and virology.
- First, the MS is way too long.
R: The manuscript is too long because it includes all information, figures and controls requested by three initial reviewers of a previous version, plus three additional reviewers, and the academic editor. Because two other reviewers have approved the manuscript with these additions we cannot remove these corresponding sections. For example, another reviewer suggested that we include a whole additional analysis of hepatitis B viruses, which would make the manuscript even longer. We had to refute the suggestion by stating that the manuscript is already too long, as reviewer # 3 agrees, and cannot be substantially increased. We are concerned that if we keep adding analyses to our study we will be caught in an unending cycle of reviews/responses in which we will be testing the suggestions and every existing virus, serotypes and controls. Although this is not the case, it would seem that the reviewing process is nitpicking on details that could in fact, deviate the reader’s focus from the main tenet of the study. The main tenet is the demonstration that it is possible to detect viral genome remnants in Neanderthal sequencing raw data with corresponding statistical tests and controls for random noise. Also, one reviewer requested that in the Results a small introduction on the deamination patterns is to be included, even if redundant.
- The 34 pages contains many repetitions that should be deleted. E.g., in the Results section the methods should not be repeated, neither the results in the Discussion.
R: We believe that these are not mere repetitions, but the information is presented in Results and Discussed thereafter. This is very subjective, and we edit as most as we could. Please indicate specifically which repeats should be edited. As explained above this redundancy in Results was a request by one of the reviewers.
- The Supplementary information contains too much details (51 files, 177 Mb, one of the files is 146 Mb long).
R: We believe the reviewer meant “too many details”? These were requests by other reviewers; therefore, we must keep these files. The longest 146 Mb file is the herpesvirus megablast alignment requested by the academic editor. He requested every alignment corresponding to every tree presented.
- Scientists always have a (hard) task to filter out the most important data and final conclusions to present for the readers to keep his interest continuous.
R: We agree. But the various reviewers of this manuscript requested different data and controls and we tried to comply with all criticisms. Sometimes what one reviewer believes as superficial another finds it essential and vice-versa. The present form of the manuscript must be satisfactory to different scientists with different views and opinions. As other reviewers have accepted the manuscript, we cannot remove parts suggested by them. Several parts of this manuscript are responses to reviewers’ comments, additional analyses and controls requested by them. As we stated above (being repetitive on purpose, for emphasis): “Although this is not the case, it would seem that the reviewing process is nitpicking on details that could in fact, deviate the reader’s focus from the main tenet of the study. The main tenet is the demonstration that it is possible to detect viral genome remnants in Neanderthal sequencing raw data with corresponding statistical tests and controls for random noise.”
- A general problem is the misinterpretation of virus species concept, and thus mixing of the sequenced actual viruses with the species where they belong to. Virus species is an abstract idea about a group collecting the “adequately” similar virus types. The authors should name the virus types like human adenovirus 7 (never starting with capital) or “members of a species, e.g., that of Human mastadenovirus B [in italics]” Presently the MS applies a mixture of virus names and species names (for adenoviruses the real virus name ends with an Arabic number and the species with a capital letter). This means that they cannot “sequence a virus species”, while the human adenovirus 7 should not be written in italics (Fig. 7). (Only official virus taxon names are be written in italics, but including also the family and higher viral taxon names, like Papillomaviridae. When you speak about a “herpesvirus”, it shouldn’t start with a capital!) The origin of this mixture of names may be that GenBank applies different names for the “organism”, sometimes the types, while other times the species. Still, a uniform labelling (applying the type numbers or strain designations) for the sequences is the correct practice.
R: We corrected as suggested.
- Your final conclusion that N-adenovirus would not merit the establishment of a new species for it is absolutely correct, however, you should not cite the criteria of the obsolete 2011 ICTV book. The presently official ICTV taxonomy for AdVs (2022) requires a more than 10-15% difference in the polymerase amino acid sequence (not 5-15%) for establishing a new species.
R: We corrected the reference.
- On the phylogenetic trees, the authors should show the closest serotypes as they are the most relevant. N-adenovirus proved to be a HAdV-7 variant, thus all the other types of species Human mastadenovirus B should be shown on the tree to demonstrate their distinctness (after removing the ambiguities in your sequences): HAdV-3, -11, -14, -16, -21, -34, -35, -50, and including (at least some of) the simian (chimpanzee and gorilla AdV) members of HAdV-B. This is important as actually this lineage is presumed to be the real (continuously coevolved) gorilla lineage, just two of its ancient members switched to humans producing the B1 and the B2 “subgroups” (by now differentiated already to different serotypes) while other gorilla AdVs switched to chimpanzees. On the phylogenetic tree, you mislabelled NC010956 as B2; it is already another species namely Human mastadenovirus D. (Actually, HAdV-D is the real human lineage, which constantly coevolved with Homo sapiens). Unfortunately, you did not include any (or some/all members, i.e., at least 4 human, and about 10 chimpanzee, bonobo and gorilla AdV types) from Human mastadenovirus C in spite that sometimes you found HAdV-C members to be most similar to your Neanderthal sequences.
R: The reviewer stated above that the manuscript is too long and at the same time requests additional analyses with the inclusion of several sequences in the AdV phylogeny. The trees include the closest relatives of the Neanderthal sequences as indicated by an extensive blast search.
Some special notes:
- Adenovirus does not establish real “latency”. Reword these sentences.
R: Corrected.
- “(IV) Adenoviruses [11].” This reference is about the HAdV-C but not about the whole family.
R: Clarified in the text.
- „evolution of transmissible and nontransmissible diseases” Are there any other possibility?
R: We know that there is no third possibility here. What we meant is that this type of analysis could provide insights not only on nontransmissible diseases, such as depression, as shown in previous Neanderthal studies, but also on transmissible diseases, such as viral diseases. Nevertheless, we rewritten this sentence to avoid misinterpretation.
- „≈70% identical with chimpanzee sequences” Which chimpanzee AdV sequence? Y25? What about the simian AdV-21? It is also a chimpanzee AdV and furthermore it is exactly in the species Human mastadenovirus B. You should compare your N-adenovirus to the closest chimpanzee and gorilla AdV types.
R: Corrected. See Figure 7a.
- „family Adenoviridae, genus Mastedenovirus and in species human Adv-B or Adv-C” should be „family Adenoviridae [in italics!], genus Mastadenovirus and in Human mastadenovirus B or Human mastadenovirus C” (or HAdV-B or HAdV-C /capital V for HAdV/!) (Theoretically, you should not abbreviate a species name, but it is done generally.)
R: Corrected.
- „prototype new species”. There is no „prototype” species anymore: „candidate member for a new species”
R: We used “prototype new species” because it was the exact wording requested by another reviewer. We changed as suggested.
- „adenovirus, herpesvirus and papillomavirus infected Neanderthals” These viruses were present already in all reptiles and mammals thus it is not too probable that they would be a new infection for this human subspecies. Maximum „certain very pathogenic variants” could be new for Neanderthals.
R: We agree. Corrected as suggested.
- 50. No authors/editors, nor Publisher for this obsolete old book (9th Report”)? Is it right to cite a whole book instead of its appropriate chapters? Is it correct to cite a bookstore item instead of the original scientific publication?
R: The reviewer is correct. We corrected the reference.
- 7. The green background should be one line larger.
R: Corrected.
Comments on the Quality of English Language
- Some minore mistakes:
hypothesis .... "propose" should be "proposes"
"sequencing reads" should be rather "sequence reads"
"Human adenoviruses persistence" should be "Human adenovirus persistence"
"thresholds for taxa differentiation" > ... taxon ...
this "probabilitydependensd" on > ... depends on ...
"nucleotide similarity" > ... identity
"of theassemblies" > of the assemblies
"N-herpesvirus e N-papillomavirus" > ... and ...
R: Corrected.
This manuscript is a resubmission of an earlier submission. The following is a list of the peer review reports and author responses from that submission.
Round 1
Reviewer 1 Report
Comments and Suggestions for Authors
The manuscript by Ferreira et al concerns the in-silico detection of viral reads with similarity to human virus families with double-stranded DNA genomes, and thus the capacity to establish chronic infections, in genome collections originating from a teeth and bone sample, respectively, from two Neanderthal individuals.
Although the topic is interesting and the possibility of detecting reads belonging to chronically-infecting dsDNA viruses in ancient materials is real, the analysis presented here is not convincing at all, and should be largely redone. The discussion of the results should likewise be improved substantially.
Major comments:
-Spell virus names without caps throughout the paper. Only families, such as Herpesviridae should have caps
-Line 89: here it is stated that samples #1 and #7 from Chagyrskaya cave will be used, as described in ref [22]. However, ref [22] is only about a sample #8. Insert correct reference
-Lines 99-102: why were these two SRA’s specifically selected?
-Lines 103-104: Selecting a specific virus sequence has no value at all as a reference sequence. Virus sequences can be very variable, with many distinct genera and strains, which may exceed 10% nucleotide difference between them. The accession numbers given do not represent reference strains!
-Table 1: sample #1 and sample #7 give very different viral hits, which do hardly match those found in the further analyses. For instance, in #7 only HSV2 is found, not HSV1 detected subsequently, as shown in Fig. 1. Please comment.
-#7 was selected for further analysis, but likely teeth, such as sample #1, are a much better reservoir for viral DNA. Please discuss.
-Table 1, Legend. When >90% of the reads in the SRA are ‘dark matter’ (suppl. Fig. S11), would 0.01% of reads representing viral sequences not be very high? Although suppl. Fig. S11 states that 0.0004% of the reads are likely viral.
-Table 1: #7 contains high-risk HPV reads, which are not found in #1. An interesting finding, but not discussed.
-Figs 1-3: the sequences used are NOT reference genomes for these viruses. When using, for instance, HPV12 (Fig. 4), mainly, or solely, reads highly similar to HPV12 will be retrieved, and not HPV16 or HPV18 detected in Table 1.
-Fig. 5: Viral read length is relatively short. Interestingly, the virus with the smallest genome, HPV, shows the longest reads, contradicting what the authors state in lines 294-295.
-Fig. 7: What is shown here is how NOT to do a NJ analysis. The MegaBlast option in the Blast program is only to be used in order to get a quick overview of the relation between retrieved sequences. This is not a bona fide analysis! Why not use the phylogenetic analyses shown in the supplementary figures? It should be described what conditions were used to create the trees, and how the degenerate nucleotide codes were treated. Most phylogenetic programs will delete them, resulting in an outgroup position for the ancient genomes, as illustrated by Fig. 7 and the suppl. Figures S4-S9. It would be very unlikely that, should the relatively recently deceased Neanderthals studied here be infected by modern humans, their viruses would cluster completely outside the known human virus clusters, as is stated in the abstract.
-Lines 270-280: delete, as these analyses are too superficial and completely meaningless. For instance, the closest relative to the N-Herpesvirus, KX265031.1, turns out to be an artificial recombinant isolated from mice infected with two HSV1 strains in the lab.
-Lines 281-289: Because of the ambiguous, degenerate positions identified in the ancient reads, an analysis of amino acid substitutions here is largely useless, and should be deleted.
- Line 289: I was unable to open the VCF files.
- The analyses described on pages 12-14 are not summarized, or discussed at all. What do the results imply? Significance of the findings?
- Fig. 10, legend: mtDNA is not a chromosome
- Discussion: most can be deleted, as the analyses were done poorly, and no conclusions can be drawn. Instead of discussing the types of DNA modifications seen in ancient DNA, it would be better to explain how such mutations can be identified, and handled in comparisons with modern DNA. There are many papers presenting and discussing the evolution of pathogens, including viruses such as HBV, in an excellent way. Use those to find out how the research is done!
- Lines 384-385: Explain how adenoviral DNA can turn up in bone
Conclusions: rewrite completely
Overall: It is unlikely that the findings obtained here are genuine, as the results from the first analysis are not replicated by the subsequent analyses; viral genus and genotypes differ substantially. The reference genomes used are not reference genomes, but represent individual isolates with limited coverage of actual viral variation. And it is likely that herpesvirus infections by the known human herpesviruses predate the human-Neanderthal split, see also for instance DOI: 10.1093/ve/vex026.
Comments on the Quality of English LanguageThere are many grammatical mistakes in the manuscript, such as line 10 viral DNA genomes, line 14-15, deamination patterns typical for ancient DNA, line 32 'have been', not 'were'. etc. Check throughout
Author Response
Reviewer #1
The manuscript by Ferreira et al concerns the in-silico detection of viral reads with similarity to human virus families with double-stranded DNA genomes, and thus the capacity to establish chronic infections, in genome collections originating from a teeth and bone sample, respectively, from two Neanderthal individuals.
Although the topic is interesting and the possibility of detecting reads belonging to chronically-infecting dsDNA viruses in ancient materials is real, the analysis presented here is not convincing at all, and should be largely redone. The discussion of the results should likewise be improved substantially.
R: Although we thank the reviewer for taking the time to evaluate our study, we found issues in the assessment which we will detail below.
Major comments:
-Spell virus names without caps throughout the paper. Only families, such as Herpesviridae should have caps
R: Corrected.
-Line 89: here it is stated that samples #1 and #7 from Chagyrskaya cave will be used, as described in ref [22]. However, ref [22] is only about a sample #8. Insert correct reference
R: Corrected.
-Lines 99-102: why were these two SRA’s specifically selected?
R: As we stated in the beginning of our Results section: An initial taxonomic analysis of sequencing reads, based on MinHash-based k-mers, (STAT, https://www.ncbi.nlm.nih.gov/sra), revealed that reads of human pathogenic DNA viruses such as herpesviruses, papillomaviruses, and adenoviruses, are present in Neanderthal sequencing runs in in approximately 0.01% reads (Chagyrskaya 1 and 7) (Table 1). Chagyrskaya 1, comprises 11 runs with 2.96x109 bases (3.25 Gb of data). To test if the reads indicated as viral in the SRA analysis of Neanderthal reads, we analyzed in more detail these reads by mapping to extant viral genomes.
Because of higher coverage of Neanderthal genome Chagyrskaya 7 (4.9 fold coverage) it was used for reference mapping using BBMap, an assembler optimized for short reads. Before processing, the data comprised 128 runs with 152.71 x 109 bases (170.21 Gb of data). Other SRAs of Chagyrskaya Cave were not tested but will be addressed in a future study.
-Lines 103-104: Selecting a specific virus sequence has no value at all as a reference sequence. Virus sequences can be very variable, with many distinct genera and strains, which may exceed 10% nucleotide difference between them. The accession numbers given do not represent reference strains!
R: We did assemblies with NCBI RefSeq sequences for all three viruses with all their RefSeq counterparts. This is included in the revised version.
-Table 1: sample #1 and sample #7 give very different viral hits, which do hardly match those found in the further analyses. For instance, in #7 only HSV2 is found, not HSV1 detected subsequently, as shown in Fig. 1. Please comment.
R: Sample 1 and sample 7 are different individuals. Therefore, they might be infected with different viruses and viral strains, so it is not unexpected that their pattern of pathogens differs. This is an epidemiological aspect that is beyond the scope of this study.
-#7 was selected for further analysis, but likely teeth, such as sample #1, are a much better reservoir for viral DNA. Please discuss.
R: Sample 7 has much more sequencing data and coverage as compared to sample 1. We explain this in the text. Therefore, if we were looking for vestigial viral DNA embedded in these Neanderthal samples, and assemble the viral genomes with better coverage, we thought we should look in a sample with more sequencing data.
-Table 1, Legend. When >90% of the reads in the SRA are ‘dark matter’ (suppl. Fig. S11), would 0.01% of reads representing viral sequences not be very high? Although suppl. Fig. S11 states that 0.0004% of the reads are likely viral.
R: 0.01% of identified reads are likely viral whereas 0.0004% of total reads (including unidentified) are likely viral.
-Table 1: #7 contains high-risk HPV reads, which are not found in #1. An interesting finding, but not discussed.
R: Although we agree that this is interesting, we prefer to save this for future study.
-Figs 1-3: the sequences used are NOT reference genomes for these viruses. When using, for instance, HPV12 (Fig. 4), mainly, or solely, reads highly similar to HPV12 will be retrieved, and not HPV16 or HPV18 detected in Table 1.
R: We use the term “reference sequence” as “assembly reference sequence”, or template as some researchers call, not “taxonomic reference sequence”. We made this distinction explicit in the revised version and used NCBI RefSeq sequences as discussed in additional analyses.
-Fig. 5: Viral read length is relatively short. Interestingly, the virus with the smallest genome, HPV, shows the longest reads, contradicting what the authors state in lines 294-295.
R: The stamen in lines 294-295 was removed. Nevertheless, the reviewer is exaggerating our argument. We mean that all viral sequences as compared to host sequences are substantially smaller by at least 6 orders of magnitude and more importantly the viral sequences are not packaged as chromatin. The mean difference between HPV and adenovirus and herpesvirus are not significant. Also, the effect seen in Figure 5 is because although the distributions are similar, with a tail to the right, the reviewer must observe that the means of the distributions are equivalent but because the papillomavirus is much smaller, the peak of 15 nucleotides reaches frequency 3,000 whereas in adenovirus the peak of 15 nt is at 32,000 and herpesvirus at 190,000 and because the distribution of papillomavirus has a smaller peak the tail seem thicker, or a scale effect.
-Fig. 7: What is shown here is how NOT to do a NJ analysis. The MegaBlast option in the Blast program is only to be used in order to get a quick overview of the relation between retrieved sequences. This is not a bona fide analysis! Why not use the phylogenetic analyses shown in the supplementary figures? It should be described what conditions were used to create the trees, and how the degenerate nucleotide codes were treated. Most phylogenetic programs will delete them, resulting in an outgroup position for the ancient genomes, as illustrated by Fig. 7 and the suppl. Figures S4-S9. It would be very unlikely that, should the relatively recently deceased Neanderthals studied here be infected by modern humans, their viruses would cluster completely outside the known human virus clusters, as is stated in the abstract.
R: The reviewer is mistaken. As shown on the NCBI site (https://blast.ncbi.nlm.nih.gov/blast/treeview/treeView.cgi) in the tab “Tree Method”, there are two options: “minimum evolution” or “neighbor-joining”. The help file and references indicated in the NCBI site also clearly indicate that these are built by neighbor-joining including the reference: Saitou and Nei, Mol Biol Evol, 4:406-25, 1987 PMID: 3447015. The objective of this figure is to show that the genomes inferred from Neanderthal data are substantially different from any extant viral sequence in GenBank and therefore the possibility that it is a contamination with a modern viral sequence, either human or non-human, is highly improbable. We are not making strong taxonomic or evolutionary conclusions from this analysis. Also, the phylogenies contained in the supplementary data fully address any possible limitations of our Megablast-NJ analysis.
-Lines 270-280: delete, as these analyses are too superficial and completely meaningless. For instance, the closest relative to the N-Herpesvirus, KX265031.1, turns out to be an artificial recombinant isolated from mice infected with two HSV1 strains in the lab.
R: We disagree with the referee. We will report any findings we observe and the fact that the closest relative, as compared to natural sequences, is an artificial sequence further supports our objective to suggest that contamination with a natural extant virus is highly unlikely. We maintain the text.
-Lines 281-289: Because of the ambiguous, degenerate positions identified in the ancient reads, an analysis of amino acid substitutions here is largely useless, and should be deleted.
R: Our study is basically a proof of concept, and we demonstrate that (1) the sequences inferred (“reconstructed”), (2) are not a product of contamination, (3) have all features of ancient DNA (deamination patterns) and (4) the mapping to assembly references cannot be explained by random sequence similarities. Therefore, we must report the sequences found and the degree of uncertainty associated with this type of analysis. We are not making any final claims on these data. It is an initial inference study. Future prospective testing in vitro by amplification from Neanderthal DNA with primers designed from the conserved regions here identified might help to close low coverage regions and thus confirm or not our findings. We will not remove the amino acid substitutions as shown. These are not conceptually wrong and could base interesting future analyses.
- Line 289: I was unable to open the VCF files.
R: Other reviewers did not have problems in opening a simple vcf file. Nevertheless, in addition to the simple files we provide a .txt file with the vcf file information in it with the hope that the reviewer might be able to open it and read the contents. A vcf file is a simple text file but instead of having a .txt extension it has a .vcf extension. Some systems mistaken it for a virtual card file (also .vcf). Please check that the proper program is used. Any linux, windows or mac text editor (not word processor) will properly open these files.
- The analyses described on pages 12-14 are not summarized, or discussed at all. What do the results imply? Significance of the findings?
R: Corrected.
- Fig. 10, legend: mtDNA is not a chromosome
R: As clearly stated on the NIH National Human Genome Research Institute website: (https://www.genome.gov/genetics-glossary/Mitochondrial-DNA): “Mitochondrial DNA is the circular chromosome found inside the cellular organelles called mitochondria”. The reviewer is in full disagreement with NIH and we will maintain our text in its current form.
- Discussion: most can be deleted, as the analyses were done poorly, and no conclusions can be drawn. Instead of discussing the types of DNA modifications seen in ancient DNA, it would be better to explain how such mutations can be identified, and handled in comparisons with modern DNA. There are many papers presenting and discussing the evolution of pathogens, including viruses such as HBV, in an excellent way. Use those to find out how the research is done!
R: Our study is inferential and descriptive. We are not studying the evolution of these viruses at this point. The reviewer is overinterpreting our data and text. We made an original analysis of Neanderthal data and found interesting results that cannot be explained by either contamination with modern DNA or random sequence mapping to assembly references (templates). The detailed evolution of these viruses is not our immediate objective at this point. We nevertheless thank for the encouragement and suggestion of literature.
- Lines 384-385: Explain how adenoviral DNA can turn up in bone
R: During acute adenovirus infections, DNA is found in substantial amounts in the blood of infected individuals. In fact, current diagnostic tests use detection of adenoviral DNA in blood (see: https://emedicine.medscape.com/article/211738-workup?form=fpf#c3) where it reads: “Detection of adenovirus DNA in the blood by quantitative PCR is increasingly utilized for the evaluation of adenovirus infections in immunocompromised patients. Studies have demonstrated an association between rising or high-level viremia and the risk of both invasive disease and mortality. Reference: doi: 10.1128/CMR.00052-07”. As is widely known, bones are irrigated by blood (see Figure 1 in doi.org/10.3389/fcell.2021.635189) and contain bone marrow. Therefore, in high-level adenovirus viremia the blood within the bones contains substantial amounts of viral DNA. It is quite possible that the individual had an ongoing adenoviral infection upon death or might have died because of it. Therefore, adenoviral DNA was within the bone without having to directly infect osteocytes.
Conclusions: rewrite completely
R: Corrected.
Overall: It is unlikely that the findings obtained here are genuine, as the results from the first analysis are not replicated by the subsequent analyses; viral genus and genotypes differ substantially. The reference genomes used are not reference genomes, but represent individual isolates with limited coverage of actual viral variation. And it is likely that herpesvirus infections by the known human herpesviruses predate the human-Neanderthal split, see also for instance DOI: 10.1093/ve/vex026.
R: We clarified the point on reference genomes. We also did novel analyses with NCBI RefSeq data that confirm our data and conclusions. Our study is not on taxonomy or evolution of these viruses and the reviewer is overemphasizing this aspect which is neither central nor a purpose of the study. The claim that the results of the first analysis are not replicated by the second analysis is incorrect. Which two analyses is the reviewer referring to? Be specific. The initial assessment of our study wrongly claimed that it was due to random similarities between short sequencing reads and the assembly reference sequences. We show that this is not the case. The reads that assemble to adenovirus, herpesvirus and papillomavirus give a substantially higher coverage as compared to random reference sequences (random noise) and do not align at all to human genome sequences as suggested in the initial assessment of our study. Therefore, the initial reviewer’s evaluation of our study (the very first version from early 2023) was incorrect and our data and conclusions hold.
Comments on the Quality of English Language
There are many grammatical mistakes in the manuscript, such as line 10 viral DNA genomes, line 14-15, deamination patterns typical for ancient DNA, line 32 'have been', not 'were'. etc. Check throughout
R: We verified the text accordingly.

Reviewer 2 Report
Comments and Suggestions for Authors
Renata C. Fereira et al. looked for viral sequences in sequencing data from Neanderthal bones taken from a Russian cave. Sequences homologous to those of contemporary DNA viruses (reference viruses) known to persist in patients at low level after the acute phases of infection were found. In particular sequences homologous to Adenovirus, Herpesvirus and Papillomavirus were detected in the “non-human” sequencing data.
The sequencing reads used to reconstruct viral sequences are derived from short nucleic acid fragments exhibiting the characteristic substitutions found at the extremities of ancient DNA fragments. Such viral sequences cannot be reconstructed from artificially generated random sequencing reads. Comparisons between ancient and modern polypeptidic sequences showed that most differences are located in regions encoding viral proteins known for their functional biological interactions with host proteins.
These results are surprising and quite interesting. However, it is unclear whether the authors i) first searched for many viral sequences present in the sequence data and then discovered the three viruses described in this study or ii) whether they only searched for the sequences of these viruses known to establish persistency in infected individuals (i.e. Herpesvirus, adenovirus and papillomavirus). If the authors, proceeded according to ii) as understood by this referee, it is then surprising that they found all viruses they looked for. It is then regrettable that they did not introduce as a control the search for viral sequences from a non-persistent double-stranded DNA virus. Did these few Neanderthals keep track of all known DNA viruses?
Minor comments:
1) Have viral sequences (retroviral sequences?) already been described in these Neanderthal sequencing data, if so, mention them in the introduction.
2) Supplementary Data: sequences in fasta format are not readable by common readers (another format can be used e.g. “simple text”).
3) Supplementary Data: figure legends would be helpful.
Comments on the Quality of English Language
No comments
Author Response
Reviewer #2
Comments and Suggestions for Authors
Renata C. Fereira et al. looked for viral sequences in sequencing data from Neanderthal bones taken from a Russian cave. Sequences homologous to those of contemporary DNA viruses (reference viruses) known to persist in patients at low level after the acute phases of infection were found. In particular sequences homologous to Adenovirus, Herpesvirus and Papillomavirus were detected in the “non-human” sequencing data.
R: There is a discussion on whether Neanderthals are humans or not. Since technically Neanderthals are a subspecies (Homo sapiens neanderthalensis as opposed to Homo sapiens sapiens) we refrain from stating that Neanderthal data is “non-human”.
The sequencing reads used to reconstruct viral sequences are derived from short nucleic acid fragments exhibiting the characteristic substitutions found at the extremities of ancient DNA fragments. Such viral sequences cannot be reconstructed from artificially generated random sequencing reads. Comparisons between ancient and modern polypeptidic sequences showed that most differences are located in regions encoding viral proteins known for their functional biological interactions with host proteins.
These results are surprising and quite interesting. However, it is unclear whether the authors i) first searched for many viral sequences present in the sequence data and then discovered the three viruses described in this study or ii) whether they only searched for the sequences of these viruses known to establish persistency in infected individuals (i.e. Herpesvirus, adenovirus and papillomavirus). If the authors, proceeded according to ii) as understood by this referee, it is then surprising that they found all viruses they looked for. It is then regrettable that they did not introduce as a control the search for viral sequences from a non-persistent double-stranded DNA virus. Did these few Neanderthals keep track of all known DNA viruses?
R: As a matter of fact, we tested for RNA viruses for negative control and no reads mapped. Also, our negative control with random mock sequences show that the similarities here observed cannot be explained by random similarities due to short reads. This is an initial study and proof of concept. We plan to expand to more Neanderthal samples and tests for non-persistent DNA viruses. We thank you for the suggestion. For example, adenovirus produces persistent infections in individuals with problems in the immune system and in healthy individuals the DNA is detected in blood during high viremic acute stages. Although such controls would be interesting, we never know if the individual from which the samples were taken were suffering from an acute infection with the nonpersistent type upon death. Although the negative result is interesting the positive result must just suggest acute infection.
Minor comments:
1) Have viral sequences (retroviral sequences?) already been described in these Neanderthal sequencing data, if so, mention them in the introduction.
R: These viral sequences have not been searched in these data as far as we know.
2) Supplementary Data: sequences in fasta format are not readable by common readers (another format can be used e.g. “simple text”).
R: Corrected in the revised version in which we provide several file formats. Please pay attention to file extensions.
3) Supplementary Data: figure legends would be helpful.
R: Thank you for the suggestion. Legends were included in the Supplementary Data.
Comments on the Quality of English Language
No comments

Reviewer 3 Report
Comments and Suggestions for Authors
Overall I think this is an exciting topic to think about however the steps you took in the analysis were not clearly laid out in this paper, I could understand what you were doing and the methods are very detailed but I was often not sure WHY you were testing what was being tested in each step. The introduction and abstract lead the reader to believe there may be some evidence of mutations in these DNA viruses that could indicate an impact on Neanderthal survival but this is never mentioned or linked to any of the analyses performed in this work. It's almost like the introduction and discussion are one paper and the results and methods are another!
Figures 2-4 there is an olive green/dark green/red bar that is to indicate the identity of the mapped reads compared to the reference but there is not a key to indicate what olive green vs dark green vs red shows about identity.. is red high identity or low identity? Also this spectrum of colors will be very difficult for those that are color blind to distinguish.
Figure 5 The read length distribution is poorly displayed - I have to take your word for it that there are reads that are 130bp in length as that part of the plot has bars that are not visible to the reader (supposedly) an inset of these plots would be useful to show the full spectrum of read length distribution .. if this is important (more on this later)
Figure 6 and paragraph on lines 224-231, what is deamination , what would the distribution of deaminated basepairs look like in a modern vs ancient sample. This should really be introduced better this is a short paragraph and there is a little explanation in the last sentence but I had to look this up for myself outside this paper.. Please structure your paragraphs better where you introduce the null hypothesis, the experimental design (analysis), and then the result. This paragraph starts with the result and ends with a conclusion but has no set up! It would be nice if you could show that these deamination patterns are unique to these viruses and are not present in reads that were binned as Bacteria and Archaea which likely represent sample contamination from sample exposure over time.
Figure 7 there is no explanation of the green vs grey dots or of the green triangles. Please describe the figure better in the legend - if you add it to the picture there should be AT LEAST a key but also an explanation in the figure legend.
Line 293-294: "Also, the average read size is smaller than segments mapped to the host genome. Viral genomes being significantly smaller than the hosts will be proportionately more affected by degradation. " You have no citation for this statement and I have never heard this before and do not understand why you think this would be true.. why would genome size impact the rate of degradation? Also if these were latent infections the genome would have been embedded in the host genome and thus the "size" of the genome would have been the same as the host DNA, if these were active infections the genome size could play a role, though I would find it hard to believe that all copies of the virus popped out of the host genome there would be both small active copies of the genome and larger latent copies embedded in the host chromosome. If you want to keep this sentence (and figure 5) you should find a reference that backs up this claim AND show the distribution of read sizes mapped to the host genome.
Figure 8 again no key for the identity bar. Also how are you distinguishing deamination from mutation? That method is not clearly laid out, perhaps this was done with the consensus sequence. If that is the case the threshold for inclusion as a mutation vs DNA damage should be clearly outlined.
Figure 9 would be more effective if I could directly compare mapping stats between the actual viral sequence and the randomly generated reference sequence. Perhaps a table would be a more effective way of demonstrating your point here.
Line323-339 I do not understand the rationale behind extracting reads that mapped to the random reference and mapping them to the hg38 assembly- please clearly explain this rational in the paragraph. Again each paragraph should CLEARLY outline the null hypothesis, experimental design, steps of the analysis, results, and lastly conclusions from the results.
Figure 11: I do not understand the rational for this analysis. Please clearly explain the null hypothesis being tested, the experimental design, the steps of the analysis, the results, and lastly the conclusions from the results.
Was the Chagyrskaya 1 sample used at all in this work? It is mentioned in the introduction and discussion but not in the body of the paper and I don't think there are any data in any of the figures? If it is not used or shown in any of the figures perhaps leave it out of the introduction and discussion.
Line 370-375 the statement made earlier about the fragmentation of viruses being faster than the host because of the small genome size, if that is true would you expect to be able to amplify repetitive regions with PCR targeting of conserved sites. IF these are degrading very rapidly I would expect a big old smear on the gel.
Line 383 -389 The paragraph starts by saying that this is a spinal sample which is where we would expect to find HSV particles and then goes on to explain that adenovirus and papilloma virus would be found elsewhere.. what is the conclusion here this is just a dump of information. Is the coverage lower for these latter two viruses because the sample is not the target tissue of an active infection? Is the coverage the same between all three.. I'm not sure how the location of the active infection of the latter two not being the site that was sampled is connected to this story or supports the conclusion that there was a a high viral load in this individual when they died.
Line 395 were there any reads that mapped to the viral references that did NOT show the pattern of deamination that one would expect? Is it deamination of the ends or in the center of the read as well that would be expected ?
The discussion of deamination on lines 406-428 is great and should be used in the results section to introduce deamination results. i.e. null hypothesis is translation vs transition rates in modern DNA compared to ancient DNA.
Line 472-473 I'm not clear on what part of this work constitutes "We provide provocative new evidence on the open question regarding the impact of viruses in the extinction of an ancestral human population ". IMHO the further analysis suggested in lines 477-479 would be provocative new evidence. In this paper you detected viral reads, assembled them, tested that they were indeed ancient and likely present at the time of the hosts death, and then inferred the sequencing. Certainly cool work but not sure this is proactive new evidence as the association of these DNA viruses with Neanderthals was already known - however this might be the first time DNA viruses have been reconstructed from Neanderthal genomes.
Author Response
Reviewer #3
Comments and Suggestions for Authors
Overall I think this is an exciting topic to think about however the steps you took in the analysis were not clearly laid out in this paper, I could understand what you were doing and the methods are very detailed but I was often not sure WHY you were testing what was being tested in each step. The introduction and abstract lead the reader to believe there may be some evidence of mutations in these DNA viruses that could indicate an impact on Neanderthal survival but this is never mentioned or linked to any of the analyses performed in this work. It's almost like the introduction and discussion are one paper and the results and methods are another!
R: In the introduction we tried to put in context the motivation of why we believe it is important to search for remnants of viral genomes in Neanderthal samples, since these viruses might be involved in their extinction as proposed by Wolff and Greenwood (doi:10.1016/j.mehy.2010.01.048). However, our study is a proof of concept. We demonstrate that vestigial viral genomes can be detected in Neanderthal reads and perform three tests, with random reference sequences (not random reads), to show that the coverage level of the mapping coverage is above the expected by random noise using reads of the lengths between 12 and 18 bases.
Figures 2-4 there is an olive green/dark green/red bar that is to indicate the identity of the mapped reads compared to the reference but there is not a key to indicate what olive green vs dark green vs red shows about identity.. is red high identity or low identity? Also this spectrum of colors will be very difficult for those that are color blind to distinguish.
R: Corrected.
Figure 5 The read length distribution is poorly displayed - I have to take your word for it that there are reads that are 130bp in length as that part of the plot has bars that are not visible to the reader (supposedly) an inset of these plots would be useful to show the full spectrum of read length distribution .. if this is important (more on this later)
Figure 6 and paragraph on lines 224-231, what is deamination , what would the distribution of deaminated basepairs look like in a modern vs ancient sample. This should really be introduced better this is a short paragraph and there is a little explanation in the last sentence but I had to look this up for myself outside this paper..
R: The issue of deamination is very well established in the field of ancient DNA research. We provided references and the papers associated with the program we used (MapDamage) that cover this topic. In the discussion we addressed this topic since we didn’t find it necessary to address it in the results.
Please structure your paragraphs better where you introduce the null hypothesis, the experimental design (analysis), and then the result. This paragraph starts with the result and ends with a conclusion but has no set up! It would be nice if you could show that these deamination patterns are unique to these viruses and are not present in reads that were binned as Bacteria and Archaea which likely represent sample contamination from sample exposure over time.
Figure 7 there is no explanation of the green vs grey dots or of the green triangles. Please describe the figure better in the legend - if you add it to the picture there should be AT LEAST a key but also an explanation in the figure legend.
R: Corrected.
Line 293-294: "Also, the average read size is smaller than segments mapped to the host genome. Viral genomes being significantly smaller than the hosts will be proportionately more affected by degradation. " You have no citation for this statement and I have never heard this before and do not understand why you think this would be true.. why would genome size impact the rate of degradation? Also if these were latent infections the genome would have been embedded in the host genome and thus the "size" of the genome would have been the same as the host DNA, if these were active infections the genome size could play a role, though I would find it hard to believe that all copies of the virus popped out of the host genome there would be both small active copies of the genome and larger latent copies embedded in the host chromosome. If you want to keep this sentence (and figure 5) you should find a reference that backs up this claim AND show the distribution of read sizes mapped to the host genome.
R: We removed this statement.
Figure 8 again no key for the identity bar. Also how are you distinguishing deamination from mutation? That method is not clearly laid out, perhaps this was done with the consensus sequence. If that is the case the threshold for inclusion as a mutation vs DNA damage should be clearly outlined.
R: The distinction between deamination and mutation is made by the method implemented in mapDamage program and discussed in
Figure 9 would be more effective if I could directly compare mapping stats between the actual viral sequence and the randomly generated reference sequence. Perhaps a table would be a more effective way of demonstrating your point here.
R: Corrected. Table 3 was included.
Line323-339 I do not understand the rationale behind extracting reads that mapped to the random reference and mapping them to the hg38 assembly- please clearly explain this rational in the paragraph. Again each paragraph should CLEARLY outline the null hypothesis, experimental design, steps of the analysis, results, and lastly conclusions from the results.
R: It is explained in the text in the revised version.
Figure 11: I do not understand the rational for this analysis. Please clearly explain the null hypothesis being tested, the experimental design, the steps of the analysis, the results, and lastly the conclusions from the results.
R: It is explained in the revised version.
Was the Chagyrskaya 1 sample used at all in this work? It is mentioned in the introduction and discussion but not in the body of the paper and I don't think there are any data in any of the figures? If it is not used or shown in any of the figures perhaps leave it out of the introduction and discussion.
R: The binning data of Chagyrskaya 1 was analysed and the same viruses were present but because the sequencing coverage was small (0.2x coverage) the assemblies by mapping were done in Chagyrskaya 7 (4.9x coverage).
Line 370-375 the statement made earlier about the fragmentation of viruses being faster than the host because of the small genome size, if that is true would you expect to be able to amplify repetitive regions with PCR targeting of conserved sites. IF these are degrading very rapidly I would expect a big old smear on the gel.
R: The statement was removed.
Line 383 -389 The paragraph starts by saying that this is a spinal sample which is where we would expect to find HSV particles and then goes on to explain that adenovirus and papilloma virus would be found elsewhere.. what is the conclusion here this is just a dump of information. Is the coverage lower for these latter two viruses because the sample is not the target tissue of an active infection?
R: This is a possibility, but in this case it is conjectural.
Is the coverage the same between all three.. I'm not sure how the location of the active infection of the latter two not being the site that was sampled is connected to this story or supports the conclusion that there was a a high viral load in this individual when they died.
R: We don’t know if upon death the individual has an acute infection with any of these viruses. We only made the point that the HSV sample was extracted from a part at least approximately consistent with the infection site. If this individual had an active infection the viremia should be high and the presence of viral DNA in the blood is significant and therefore distributed in various tissues. In the case of adenovirus, the viral DNA is readily detected in the blood in immunocompromised patients.
Line 395 were there any reads that mapped to the viral references that did NOT show the pattern of deamination that one would expect?
R: If there are such reads, they are below the expected by the likelihood model implemented in mapDamage program.
Is it deamination of the ends or in the center of the read that as well would be expected ?
R: The patterns match very well and are discussed in the Results section page 5 and in Discussion section on page 9.
The discussion of deamination on lines 406-428 is great and should be used in the results section to introduce deamination results. i.e. null hypothesis is translation vs transition rates in modern DNA compared to ancient DNA.
R: Corrected.
Line 472-473 I'm not clear on what part of this work constitutes "We provide provocative new evidence on the open question regarding the impact of viruses in the extinction of an ancestral human population ". IMHO the further analysis suggested in lines 477-479 would be provocative new evidence. In this paper you detected viral reads, assembled them, tested that they were indeed ancient and likely present at the time of the hosts death, and then inferred the sequencing. Certainly cool work but not sure this is proactive new evidence as the association of these DNA viruses with Neanderthals was already known - however this might be the first time DNA viruses have been reconstructed from Neanderthal genomes.
R: The conclusion was rewritten.
Round 2
Reviewer 1 Report
Comments and Suggestions for Authors
In the revised version, some improvements have been made, but overall, the study is still confusing. The authors refused to incorporate many suggested improvements (R: We disagree with the referee. We will report any findings we observe and the fact that the closest relative, as compared to natural sequences, is an artificial sequence further supports our objective to suggest that contamination with a natural extant virus is highly unlikely. We maintain the text.), or misinterpreted the comment, such as:
-Table 1: sample #1 and sample #7 give very different viral hits, which do hardly match those found in the further analyses. For instance, in #7 only HSV2 is found, not HSV1 detected subsequently, as shown in Fig. 1. Please comment.
R: Sample 1 and sample 7 are different individuals. Therefore, they might be infected with different viruses and viral strains, so it is not unexpected that their pattern of pathogens differs. This is an epidemiological aspect that is beyond the scope of this study.
This is not what was meant, it is not about a diference between sample #1 and #7, but about sample #7, which gives hits with HSV2 in Table 1, but with HSV1 in Fig. 1. Which are related, yet still very different viruses.
Furthermore, the authors state that they have used viral consensus sequences to align the N-viruses with (page 12), but in Fig. 2-4, and the legends, it is still stated that the reads have been aligned to a single accession number for each virus.
Next, in Fig. 7, the authors use the NCBI-drawn trees, which I do not understand, as all other phylogenetic analyses have been done (suppl. information). In the description of the results, it is stated that all N-viruses were highly divergent, and clustered outside their reference set. Which is clearly not true for N-papillomavirus (Fig. 7C). And, how was ancient DNA damage, resulting in substitutions, affecting the results? DNA damage, showing patterns expected for true ancient reads was substantial, as was shown in Fig 6, and 2-4. Such mutations could easily affect the phylogenetic analyses, and the conclusion that N-viruses are highly divergent.
Comments on the Quality of English LanguageThere are still many grammatical errors throughout the manuscript.
Confusing is also for instance a sentence on page 10: "In Figure 6 B. 1F, in the Herpesvirus assembly, although C-T deamination is more prevalent in the termini the C-T is more abundant in central regions than other changes"
Author Response
Response to Reviewer #1
In the revised version, some improvements have been made, but overall, the study is still confusing. The authors refused to incorporate many suggested improvements (R: We disagree with the referee. We will report any findings we observe and the fact that the closest relative, as compared to natural sequences, is an artificial sequence further supports our objective to suggest that contamination with a natural extant virus is highly unlikely. We maintain the text.),
R: We just disagree with this specific point and accepted all other relevant suggestions. We thank the reviewer for the careful attention to the details and calling up our attentions for errors.
or misinterpreted the comment, such as:
-Table 1: sample #1 and sample #7 give very different viral hits, which do hardly match those found in the further analyses. For instance, in #7 only HSV2 is found, not HSV1 detected subsequently, as shown in Fig. 1. Please comment. R: Sample 1 and sample 7 are different individuals. Therefore, they might be infected with different viruses and viral strains, so it is not unexpected that their pattern of pathogens differs. This is an epidemiological aspect that is beyond the scope of this study.
This is not what was meant, it is not about a diference between sample #1 and #7, but about sample #7, which gives hits with HSV2 in Table 1, but with HSV1 in Fig. 1. Which are related, yet still very different viruses.
R: The reviewer is correct. The point is now clear and has been addressed in the new version with analyses shown in Figures 13 and 14 and the corresponding text in sections "Results" and "Discussion".
Furthermore, the authors state that they have used viral consensus sequences to align the N-viruses with (page 12), but in Fig. 2-4, and the legends, it is still stated that the reads have been aligned to a single accession number for each virus.
R: The details (zoom) of the assemblies shown are from tha file containing the reads mapped to different references (templates) but the program just shows on each BAM file the reads mapped to each individual reference and not all of them at the same time. The consensus sequence of the assembly is not a consensus of several references but the consensus formed between the reference sequence (single) and all reads mapped to it. The consensus sequence is in general the majority of base states using also a quality criteria.
Next, in Fig. 7, the authors use the NCBI-drawn trees, which I do not understand, as all other phylogenetic analyses have been done (suppl. information). In the description of the results, it is stated that all N-viruses were highly divergent, and clustered outside their reference set. Which is clearly not true for N-papillomavirus (Fig. 7C). And, how was ancient DNA damage, resulting in substitutions, affecting the results? DNA damage, showing patterns expected for true ancient reads was substantial, as was shown in Fig 6, and 2-4. Such mutations could easily affect the phylogenetic analyses, and the conclusion that N-viruses are highly divergent.
R: This has been corrected and the Bayesian trees are in Figure 7 and the MegaBlast NJ have been moved to the supplementary data. Both Figure 7 and the corresponding text address the excelent point made by the reviewer.
There are still many grammatical errors throughout the manuscript.
R: We scanned thoroughly the manuscript with the spelling and grammar check of microsoft word and no errors according to this method remains. Would you be so kind as to indicate the grammatical errors, specifically, that are present in the manuscript?
Confusing is also for instance a sentence on page 10: "In Figure 6 B. 1F, in the Herpesvirus assembly, although C-T deamination is more prevalent in the termini the C-T is more abundant in central regions than other changes"
R: This has been corrected in the revised manuscript. Thanks.
Reviewer 2 Report
Comments and Suggestions for Authors
This reviewer is sorry to note that the only negative result (control) in this study is for RNA viruses. RNA is a fragile molecule that cannot be compared to DNA in terms of long-term resistance. Research for RNA viruses is not included in this article.
Is it possible to find a single DNA virus that did not infect (that is not associated with) this very small population of Neanderthals? The authors have not yet answered this question. If the answer is negative, what are these ancient viral DNA sequences (modified by deamination) derived from?
In any case, a relevant negative control (e.g. non-persistent DNA viruses) is required for this “proof of concept” study.
Comments on the Quality of English LanguageMinor revision
Author Response
Response to Reviewer #2
This reviewer is sorry to note that the only negative result (control) in this study is for RNA viruses. RNA is a fragile molecule that cannot be compared to DNA in terms of long-term resistance. Research for RNA viruses is not included in this article.
R: The reviewer is correct in stating that RNA is fragile and not a proper control.
Is it possible to find a single DNA virus that did not infect (that is not associated with) this very small population of Neanderthals? The authors have not yet answered this question. If the answer is negative, what are these ancient viral DNA sequences (modified by deamination) derived from?
R: Most sequencing reads in these Neanderthal data are classified "Dark Matter" by NCBI. They use a binning method to classify these sequences (Please see our supplementary data S10 and S11). The nature of reads that do not map to any references is stil a matter of debate and are beyond the scope of our present study. For a discussion please see: https://journals.plos.org/plosone/article?id=10.1371/journal.pone.0018011.
In any case, a relevant negative control (e.g. non-persistent DNA viruses) is required for this “proof of concept” study.
R: We included a nopersistant DNA virus as control in Figure 12 and the corresponding text in the revised version.
Reviewer 3 Report
Comments and Suggestions for Authors
The changes that were made addressed the comments and concerns that I previously pointed out. This is a stronger paper for those changes IMHO.
Author Response
We thank the reviwer for the time spent to evaluate our manuscript and the suggestions.
Round 3
Reviewer 1 Report
Comments and Suggestions for Authors
The second revision of the paper by Ferreira et al shows significant improvements, especially by the additional analyses the authors have done, such as adding parvovirus B19 to the query set, an acute DNA virus for which reads would not be expected in the ancient sample, and by tackling the HSV-1/HSV-2 comment. In addition, the phylogenetic analyses improved by now showing the Bayesian trees (Fig. 7). However, it would have been interesting to see how the ancient virus genomes behave with a different reference set, as it appears that only close relatives have been used. To show how similar -or different- Neanderthal viruses are from the human ones, an analysis with virus genomes from other primates, such as chimpanzees, would be very helpful.
As the revision did not have track changes, it was difficult to assess exactly what has been changed, but overall the writing seems more cautious, which is good.
Some other points:
- Add to the methods how ancient DNA damage was handled in the phylogenetic analyses. Were degenerate codes removed? Please state the effective length of the aligned sequences used per tree analysis
- section 3.6: the amino acid changes mentioned cannot easily be retrieved from the mentioned supplementary files, as those are nucleotide sequences. And, how reliable are the changes mentioned, as the consensus sequences show many degenerate codes likely due to DNA damage?
- To the legends of Fig. 1-4, add that an example of the alignments to the references is shown. Otherwise, readers will be confused, like me, believing that the analysis was done using a single reference sequence
- Discuss the chances of finding three DNA virus genomes in a single sample. It would make sense to detect the three virus genomes in a single individual, but in a single sample it should be quite rare, I suppose.
Comments on the Quality of English LanguageSome minor adjustments should be made, which will be done during the production process, I trust
Author Response
The second revision of the paper by Ferreira et al shows significant improvements, especially by the additional analyses the authors have done, such as adding parvovirus B19 to the query set, an acute DNA virus for which reads would not be expected in the ancient sample, and by tackling the HSV-1/HSV-2 comment. In addition, the phylogenetic analyses improved by now showing the Bayesian trees (Fig. 7).
R: We thank the reviewer for the constructive criticism that helped to improve the manuscript.
However, it would have been interesting to see how the ancient virus genomes behave with a different reference set, as it appears that only close relatives have been used. To show how similar -or different- Neanderthal viruses are from the human ones, an analysis with virus genomes from other primates, such as chimpanzees, would be very helpful.
R: A phylogenetic analysis with chimpanzee and rhesus macaque viral sequences were included in the revised form. The added and changed sections are highlighted in yellow. As stated in the manuscript: “As expected, and regarding RefSeq non-human viral sequences, when modern humans and Neanderthal viral sequences are analyzed, it is evident that chimpanzee’s viral sequences, and other primates, cluster as outgroup to the Neanderthal-modern human ingroup (Supplementary Figure S11, Supplementary Tables 10, 11 and 12). Murid sequences are a more distant outgroup relative to the primate ingroup. This is the expected pattern. We propose that a deeper, more detailed phylogenetic analysis in a future study, would require the sequencing of viral genomes directly amplified from Neanderthal DNA to close the low coverage gaps and confirm synonymous and non-synonymous SNPs to determine the intraspecific and interspecific relationships more precisely between these viral sequences.”
As the revision did not have track changes, it was difficult to assess exactly what has been changed, but overall the writing seems more cautious, which is good.
R: In the revised manuscript additions and changes have been highlighted in yellow.
Some other points:
- Add to the methods how ancient DNA damage was handled in the phylogenetic analyses. Were degenerate codes removed? Please state the effective length of the aligned sequences used per tree analysis
R: This addressed in the manuscript in Results and Discussion sections.
- section 3.6: the amino acid changes mentioned cannot easily be retrieved from the mentioned supplementary files, as those are nucleotide sequences. And, how reliable are the changes mentioned, as the consensus sequences show many degenerate codes likely due to DNA damage?
R: Supplementary Figures S10, S12, S13, S14 and S15 (with corresponding text) were added to address this point.
- To the legends of Fig. 1-4, add that an example of the alignments to the references is shown. Otherwise, readers will be confused, like me, believing that the analysis was done using a single reference sequence
R: Corrected in the revised version.
- Discuss the chances of finding three DNA virus genomes in a single sample. It would make sense to detect the three virus genomes in a single individual, but in a single sample it should be quite rare, I suppose.
R: This discussion was added at the end of the “Discussion” section just before the “Conclusions” section.
Reviewer 2 Report
Comments and Suggestions for Authors Thank you for the Parvovirus control. I am convinced it makes your work more solid. The article is now acceptable.Comments on the Quality of English Language
/
Author Response
We thank the reviewer for emphasizing the importance of the negative control. It surely improved the manuscript.